# *TL;DR*: Too Long, Do Re-weighting for Efficient LLM Reasoning Compression

## Abstract

Large Language Models (LLMs) have recently achieved remarkable progress on complex reasoning tasks by leveraging extended Chain-of-Thought (CoT) techniques. These reasoning processes can be roughly categorized into System-1 (fast and intuitive) and System-2 (slow and deliberate) paradigms. However, excessive reliance on lengthy System-2-style reasoning during inference—often producing extremely long outputs—has raised concerns about efficiency. In this work, we propose a dynamic ratio-based training pipeline that does not rely on sophisticated data annotations or interpolation between multiple models. We continuously balance the weights between the model's System-1 and System-2 data to eliminate redundant reasoning processes while preserving the model's reasoning capability. We validate our approach across multiple base models, including Deepseek-R1-Distilled Qwen models, as well as on a diverse set of benchmarks with varying difficulty levels. Our method significantly reduces the number of output tokens by nearly 40% while maintaining the accuracy of the reasoning. Our code and data are at link: https://anonymous.4open.science/r/TLDR_Review-BBE5/.

## 1 Introduction

Recent efforts have developed reasoning-oriented Large Language Models (LLMs) capable of solving complex tasks. These models progressed from System-1 to System-2 paradigms (Yu et al., 2024; Li et al., 2025). System-1 implementations, such as GPT-4o (Team, 2024b), LLaMA-3 (Team, 2024a), leverage rapid intuitive processing for immediate responses but struggle with complex reasoning tasks. In contrast, System-2 architectures such as DeepSeek-R1 (Dee, 2025) are fine-tuned with extended thinking chains to promote deliberate analysis through iterative self-assessment, error mitigation, and verification, albeit facing challenges related to redundancy.

However, reasoning LLMs often over-deliberate even on simple problems Chen et al. (2025); Wang et al. (2025), resulting in unnecessary exploration and planning that undermine their efficiency and practicality. To mitigate this issue, two broad categories of approaches have been explored: *training-free methods* (Xu et al., 2025b; Yao et al., 2025; Han et al., 2024) and *training-based methods*. Some training-free methods regulate the internal states of the model during reasoning—*e.g.*, through prompts or confidence-based techniques to compress the model outputs. Alternatively, the mainstream, exemplified by model merging (Wu et al., 2025b; Team et al., 2025) and steering (Chen et al.; Azizi et al., 2025), involves intervening in the parameters or decoding process of the reasoning LLM to produce concise solutions.

In contrast, *training-based methods* (Xia et al., 2025; Yang et al., 2025; Ma et al., 2025) primarily focus on sampling and synthesizing relatively concise reasoning paths on specified problem sets through various strategies (Xia et al., 2025; Yang et al., 2025; Ma et al., 2025). These methods involve performing reinforcement learning (Meng et al., 2024; Hou et al., 2025b; Luo et al., 2025a; Aggarwal & Welleck, 2025) or supervised fine-tuning (SFT) (Chen et al., 2025) on reasoning LLMs, enabling the model to learn to generate more concise yet correct reasoning paths. Despite their effectiveness of SFT, the training-based methods typically require careful collection of problems and precise control of the data ratio for different lengths to achieve good results, leading to a complex process of parameter tuning and data construction. For example, TOPS (Yang et al., 2025) requires pre-processing steps to manually label SFT data to construct length-sensitive models, while CoT-Valve (Ma et al., 2025) generates data by creating intermediate models through model interpolation for

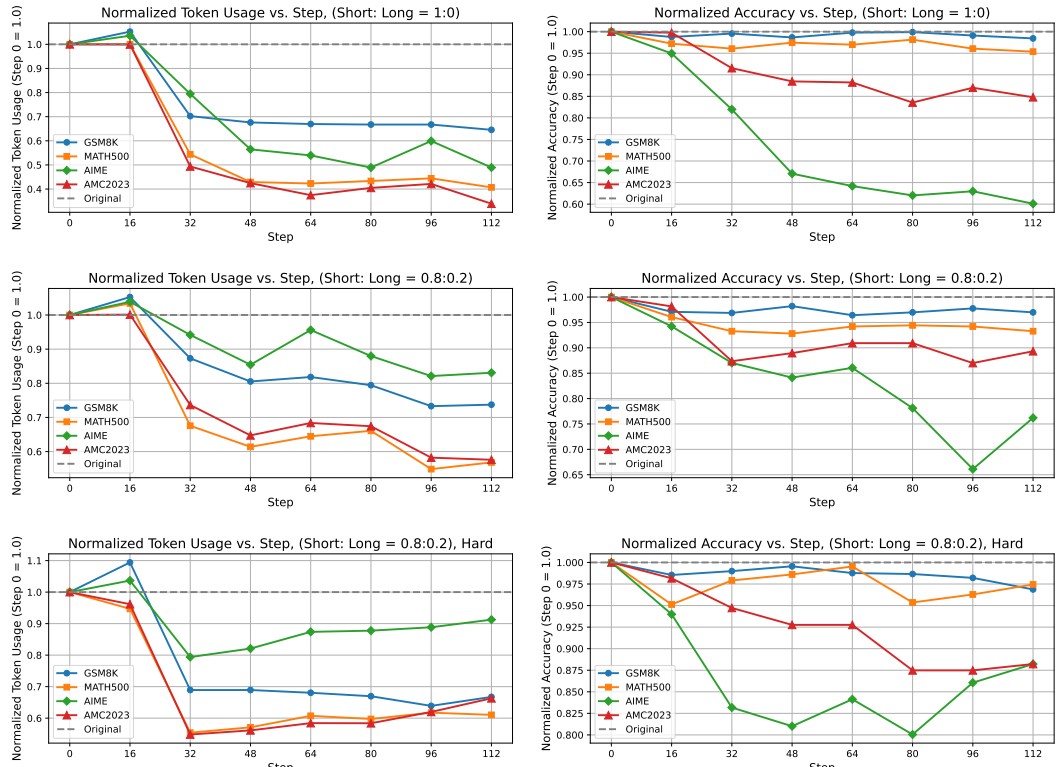

Figure 1: Impact of Combining Short CoT and Long CoT in Fixed Ratios on Thinking Compression Performance and Token Cost. We assessed the variation decay rate in output token length and accuracy on datasets of various question difficulty, spanning from GSM8K to AIME. The Normalized Token/Acc metric detail please refer to Eq. (18) and Eq. (19).

sampling. This construction process is often tedious (Yang et al., 2025), or challenging to maintain at high quality (Ma et al., 2025). RL-based methods (Aggarwal & Welleck, 2025; Hou et al., 2025a) yield more accurate and concise solutions by penalizing redundancy in System-2 reasoning, though their efficiency gains come at the cost of expensive training.

To better understand the role of different CoT instruction data in long CoT compression, we re-examine the impact of different CoT datasets on compressing long CoT models from two perspectives: problem difficulty and the length of the chain of thought with takeaways in Figure 1. Our findings indicate that two CoT subtypes have distinct effects: Long CoT on complex problems (System-2) and short CoT on easy problems (System-1) drive divergent optimization in the model's reasoning behavior. Increasing System-1 data promotes more concise reasoning but may reduce accuracy, whereas more System-2 data preserves performance on complex tasks at the cost of lower compression efficiency.

We base our approach on an intuitive motivation: when a model is thinking too long, it should reweight more intuitive reasoning paths to simplify the thinking process. Conversely, when the thinking is too direct, it should incorporate more slow-thinking reasoning chains to encourage deeper contemplation. We propose a dynamic **T**hinking **L**ength **D**ata **R**e-Weighting method (**TLDR**), which dynamically balances the model's complex reasoning using long CoT and efficient reasoning using short CoT data, enabling the model to eliminate redundant cognitive processes. First, we construct System-1-style short CoT data for GSM8K-like (Cobbe et al., 2021a) simple problems and System-2-style long CoT data for s1-like (Muennighoff et al., 2025) complex problems. The model begins with an initial ratio and performs reasoning compression using mixed data. After completing a compression cycle, the model re-evaluates the expected benefits of System-1 CoT data and System-2 CoT data to achieve improved performance. Specifically, and in line with intuition, System-1 CoT data can enhance efficiency, so we use an efficiency metric to measure the expected benefit of System-1 data. System-2 CoT, in contrast, improves reasoning accuracy, and we use an accuracy metric to measure the benefit of System-2 data for reasoning capability.

Compared to various methods requiring fine-tuning data with different reasoning lengths (Ma et al., 2025; Yang et al., 2025), our approach enables dynamic ratio learning by utilizing the self-sampled long CoT model and the short CoT data constructed by the original instruct/base model. As a dynamic SFT approach, our method achieves shorter training time and higher efficiency than ThinkPrune and L1, while also delivering superior performance. Through experiments on DeepSeek-Distill-7B/14B, our model achieves excellent compression results on the DeepSeek Distilled Qwen-7B/14B model models, with only a slight decrease in reasoning capability.

## 2 RELATED WORK

### 2.1 EFFICIENT SYSTEM-2 REASONING

Despite the strong generalization and reasoning abilities demonstrated by the system 2 reasoning paradigm, the auto-regressive nature of LLMs imposes a significant reasoning burden (Chen et al., 2025; Wang et al., 2025). To address this, various approaches have emerged to improve the reasoning efficiency. These methods can be broadly categorized into two types. One category focuses on building *adaptive reasoning-budget*. Within this, some training-free methods like CoD (Xu et al., 2025b) and TALE-EP (Han et al., 2024) impose budget constraints to control overall reasoning cost. Budget-sensitive models such as L1, TOPS, O1-Pruner, K1.5 (Aggarwal & Welleck, 2025; Yang et al., 2025; Luo et al., 2025b; Team et al., 2025) add length penalties during the post-training. Some work (Ma et al., 2025; Jiang et al., 2025; Yu et al., 2025) synthesizes diverse-length CoT data, while TOPS (Yang et al., 2025) samples budget-sensitive versions using a data model, and C3oT (Kang et al., 2024) compresses original LLM output. Although prior work—such as TOPS (Yang et al., 2025) and CoT-Valve (Ma et al., 2025)—has devoted considerable effort to constructing datasets with various CoT lengths to enable adaptive reasoning, few studies have compared how different reasoning-chain lengths affect a model's performance in terms of accuracy and inference length. We first constructed the short CoT data using simple problems and recorded how, as training steps increased, this subset contributed to token compression and accuracy retention across datasets of varying difficulty in math benchmarks.

### 2.2 DATA RE-WEIGHT OF LLM TRAINING

The quality and proportion of data are critical during both the pre-training and post-training phases. In the pre-training stage, data quality and proportion are primarily managed through filtering and reweighting. Pre-training data filtering, extensively studied to boost model performance and training efficiency (Liu et al., 2024; Albalak et al., 2024), typically involves steps like language filtering (Laurençon et al., 2023; Chowdhery et al., 2022), quality filtering (Raffel et al., 2023; Rae et al., 2022), content filtering (Xu et al., 2021; Longpre et al., 2023), and deduplication (Hernandez et al., 2022; Lee et al., 2022). While these methods significantly enhance corpus quality, their static nature can hinder dynamic adjustments during training, potentially discarding valuable data (Muennighoff et al., 2023) and introducing biases (Gururangan et al., 2022; Longpre et al., 2023; Dodge et al., 2021). Similarly, in the post-training stage, an appropriate proportion of data with varying characteristics is crucial for optimizing final performance. For example, DeepMath-103K generates a large volume of data with evenly distributed difficulty for training (He et al., 2025), SRPO designs a dynamic sampling approach to filter out samples that are consistently answered correctly, thereby improving inference efficiency (Zhang et al., 2025). To the best of our knowledge, we are the first to introduce a re-weighting mechanism into thinking compression. By employing simple strategies to construct short and long CoT, we enable the model to dynamically compress its reasoning process.

## 3 RETHINKING SHORT-LONG COT IN THINKING COMPRESSION

In our experiments, we constructed short CoT and long CoT based on datasets of different difficulty levels. To examine how problem difficulty influences the effectiveness of System-1 data and System-2 data, we selected two types of problems: (1) *Simple problems*, drawn from GSM8K, a benchmark of math word problem solving and (2) *Difficult problems*, drawn from S1, a deliberately curated dataset of difficult problems.

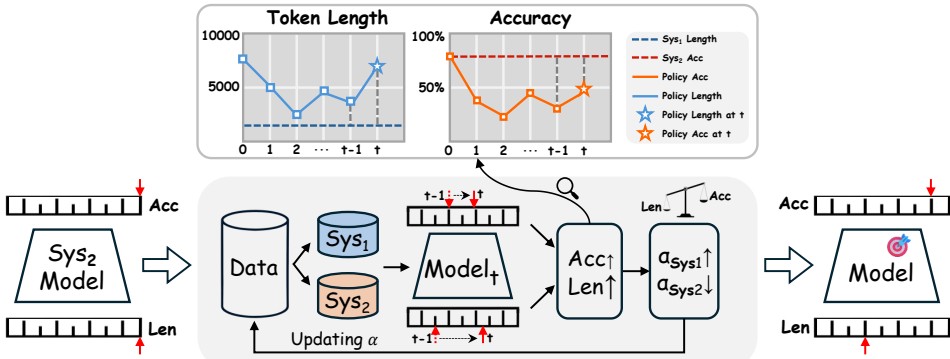

Figure 2: Overview of TLDR. Starting with a System-2 model, we iteratively update it on both Short-CoT and Long-CoT samples. Their ratios are adjusted every few steps based on the current average model accuracy and token length from the validation set, repeating until convergence.

**We find that short CoT thinking data for simple problems (System-1 data) can help compress the token usage across questions of various difficulty levels**. We leverage the short-cut solutions obtained from simple questions in GSM8K to fine-tune the model and then observe the token compression rates and accuracy drop rates across four datasets, ranging from simple to difficult: GSM8K, MATH500, AMC, and AIME. As shown in Figure 1, directly fine-tuning the long CoT model with short CoT data achieves good length compression for both simple and complex problems. We were pleasantly surprised to see that this form of length compression generalizes well across questions of all difficulty levels, and that it maintains strong performance on simple questions. However, this approach comes at a cost, as it leads to a significant decrease in reasoning ability on difficult problems. As this portion of the data is derived from intuitive CoT reasoning on simple problems, we denote it as System-1 data. It seems that directly using short CoT fine-tuning can only encourage the reasoning LLM to retain its System-1 reasoning abilities, while its ability for System-2 reasoning—slow and cautious thinking for complex problems—is largely lost.

**We find that long CoT thinking data for difficult problems (System-2 data) can help maintain the model's performance on challenging tasks**. We sample with the s1 (Muennighoff et al., 2025) like hard question prompt and then blend the System-2 data into the previous System-1 thinking dataset at a fixed short CoT *vs.* long CoT ratio: 0.8:0.2. We then observe the token compression rates and accuracy drop rates across four datasets.

It is worth noting that, by contrast, when we mix more long CoT data from simpler questions, the model still experiences a significant drop in performance on difficult questions. Refer to the middle and bottom parts of Figure 1, where we mix the long CoT sampled from challenging problems with the short CoT from simple problems. As a baseline, we also mix long CoT and short CoT from simple problems. The long CoT from difficult problems achieves lower accuracy drop rates across different datasets while maintaining comparable token compression rates. We are unable to recover the original performance simply by using long CoT data from simple questions through data replay. Similar to the deliberate reasoning characteristic of the System-2 process on difficult problems, we refer to this part of the data as System-2 data.

A key question we directly address is ***whether a direct mixing ratio of the two types of data (System-1/2 data), can be employed for post-training the long CoT model, resulting in a solution that eliminates redundancy*** without compromising performance. Based on these observations, we propose a dynamic approach aimed at identifying the optimal Thinking Compression data.

## 4    SYSTEM-1/2 DATA DYNAMIC RE-WEIGHTING

### 4.1    SYSTEM-1/2 DATA REWEIGHTING WITH RELAXED OPTIMIZATION

We formalize the thinking compression problem as an optimization task to determine the optimal ratio between System-1 and System-2 reasoning. We expect the model trained on mixed data to approach the superior performance of System-1 and System-2 in specific evaluation metrics. For model LLM and an input problem $x$, we define $\text{Token}(y) \in \mathbb{Z}^+$, $\text{Accuracy}(y) \in$ as the token length and correctness

---

**Algorithm 1** Overall Pipeline of TLDR: Data Construction and Algorithm Execution

---

**Require:** Domain data $\mathcal{D}_{\text{sys-1}}, \mathcal{D}_{\text{sys-2}}, \mathcal{D}_{\text{dev}}$; training steps $T$; batch size $b$; step size $\eta$; smoothing parameter $c \in [0, 1]$ (e.g., $c = 10^{-4}$ in our implementation). The long CoT model $\theta_{\text{long}}$ to be optimized and a reference $\theta_{\text{short}}$.
   *Step 1: Estimate the ideal upper bounds of efficiency and performance.*
   Initialize proxy weights $\theta_0$ from orginal long CoT model $\theta_l$.
   Initialize mixture weights $\alpha_0 = (1/2, 1/2)$
   Compute $\phi_{\text{sys-1,bound}}$ and $\phi_{\text{sys-2,bound}}$ using Eq. (5) and Eq. (6) using $\theta_{\text{long}}$ and $\theta_{\text{short}}$.
   *Step 2: Thinking Compression Post-Train with dynamic System-1/2 reasoning weights.*
   **for** $t = 1$ to $T$ **by** $T_d$ **do**
      Conduct SFT with the current System-1/2 data proportions and update proxy model weights $\theta_{\text{proxy}}$ using $L(\theta_{\text{proxy}}, \alpha_{t-1})$ (e.g., via Adam, Adafactor).
      Compute benefit of fine-tuning with System-1 data: $\lambda_{\text{sys-1}}$ and System-2 data $\lambda_{\text{sys-2}}$. which constitutes the approximate gradient of $\alpha_t$
      Update weights (entrywise exponential): $\alpha'_t[i] \leftarrow \alpha_{t-1}[i] \cdot \exp(-\eta \cdot \lambda_{\text{sys-i}})$
      Smooth and Renormalize: $\alpha_t[i] \leftarrow \frac{\alpha'_t[i]}{\sum_{i=1}^{k} \alpha'_t[j]}$.
   **end for**

---

of LLM output text $y$. We represent the System-1/2 ability bound as $\phi_{\text{sys-}i,\text{bound}}(y)$, in the following sections, we will abbreviate as $\phi_{\text{sys-}i,\text{bound}}(y)$

$$\min_{\theta, \, \alpha \in (0,1)} L(\theta, \alpha) = \sum_{i=1}^{2} \alpha_i \cdot \delta_i \tag{1}$$

$$\delta_i = \phi_{\text{sys-}i,\text{bound}} - \phi_{\text{sys-}i,\theta} \tag{2}$$

of which, $\phi_{\text{sys}-1,\theta}$ can be regarded as a metric for measuring the efficiency of the System-1 reasoning. $\phi_{\text{sys}-2,\theta}$ can be regarded as an accuracy metric. In this way, the overall optimization objective is to minimize the gap between the current model and the efficiency upper bound of System-1, as well as the reasoning capability upper bound of System-2, in other words, optimizing the model parameters to maximize both reasoning performance and efficiency.

$$\phi_{\text{sys}-1,\theta} = -\mathbb{E}_{y \sim \text{LLM}(x), x \in D_{\text{dev}}}[\text{Token}(y)] \tag{3}$$

$$\phi_{\text{sys}-2,\theta} = \mathbb{E}_{y \sim \text{LLM}(x), x \in D_{\text{dev}}}[\text{Accuracy}(y)] \tag{4}$$

**Setup for System-1/2 Mixed Data.** Since System-1 can provide fast and intuitive answers to simple problems, we use the short CoT model to modulate the data for the System-1 model, and the training set of basic elementary arithmetic problems from GSM8K is used as the problem dataset for System-1 data.. Meanwhile, as System-2 is designed to execute slow, logical reasoning for challenging problems, we employ the long CoT model to sample solutions from S1 (Muennighoff et al., 2025) prompts set, retaining only the correct responses. Finally, we obtain $D_{\text{sys-1}} = $ *<Simple Question, Short CoT>* instruction pairs. For the harder problems within the System-2 domain, we used the LongCoT model for sampling, resulting in a large amount of $D_{\text{sys-2}} = $ *<Hard Question, Long CoT>* instruction data.

### 4.2 SYSTEM-1/2 DATA-REWEIGHTING TUNING.

**Step 1: Estimate the ideal upper bounds of efficiency and performance.** During training, we aim to continuously adjust the ratio of System-1 and System-2 data in the post-training phase, ensuring that the model retains the reasoning capabilities of the original long CoT model while achieving the efficiency of the short CoT model. Therefore, we set the accuracy upper bound, $\phi_{sys-2,\text{bound}}$, of the model obtained through mixed training to match the accuracy of the original long CoT model, while setting the token lower bound, $\phi_{sys-1,\text{bound}}$, of the mixed model to correspond to the data lower bound of the short CoT model we constructed. The $y_s$ is the short CoT response from System-1 model with parameter $\theta_s$ and $y_l$ is the long CoT from the corresponding System-2 model parameterized by $\theta_l$.

$$\phi_{\text{sys-1, bound}} = \phi_{\text{sys-1,short}} = \phi_{\text{sys-1},\theta_s} = -\hat{\mathbb{E}}_{\text{dev}}[\text{Token}(y_s)] \tag{5}$$

$$\phi_{\text{sys-2, bound}} = \phi_{\text{sys-2,long}} = \phi_{\text{sys-2},\theta_l} = \hat{\mathbb{E}}_{\text{dev}}[\text{Accuracy}(y_l)] \tag{6}$$

**Step 2: Thinking Compression Post-Train with dynamic System-1/2 reasoning weights**    Initially, we assign an equal data ratio $\alpha_0$ to System-1 and System-2 data. Then, after every $T_d$ SFT training steps, to optimally optimize the objectives expressed by Eq. (5) and Eq. (6), we recompute the weights of System-1 and System-2 data. Specifically, we compute the gains of System-1 and System-2 data using Eq. (12) and Eq. (13), together with the previously derived upper bounds. We then solve this optimization problem using the Exponential Gradient method, with detailed derivations provided in Appendix B. Based on the solution, we determine the updated data allocation weights for the following $T_d$ steps.

$$\lambda_{\text{sys-1}} = \max \left( \frac{\phi_{\text{sys-1, bound}} - \phi_{\text{sys-1}, \theta_{\text{proxy}}}}{\phi_{\text{sys-1}, \theta_s} - \phi_{\text{sys-1}, \theta_l}}, \, 0 \right) \tag{7}$$

$$\lambda_{\text{sys-2}} = \max \left( \frac{\phi_{\text{sys-2, bound}} - \phi_{\text{sys-2}, \theta_{\text{proxy}}}}{\phi_{\text{sys-2}, \theta_l} - \phi_{\text{sys-2}, \theta_s}}, \, 0 \right) \tag{8}$$

Subsequently we determine the updated data allocation weights for the following $T_d$ steps and continue SFT proxy model. This process repeats continuously, as illustrated in Algorithm 1.

We dynamically evaluate the utility of System-1 and System-2 reasoning data's values during training, and, guided by the performance on an validation dataset, adjust the sampling ratio between the two data types in real time to optimize training effectiveness to optimize the parameter $\theta_{\text{proxy}}$. After a sufficiently large number of training steps, we select from the checkpoints that do not show degraded reasoning performance on the validation set, choosing the one with the lowest average output length.

In Figure 2, we present a concrete example of weight adjustment. After re-evaluating performance on the validation set, we readjusted the proportions of System-1 and System-2 data based on the models' output length and accuracy on the validation set. The reweighted data was then used for SFT, resulting in a model that produces fewer outputs while maintaining the original model's accuracy.

## 5 EXPERIMENTS

### 5.1 EXPERIMENTAL SETUP

**Datasets and Metrics.**    Following prior efforts, we evaluate TLDR on several widely-used benchmarks that span a broad range of difficulty levels, including ASDiv (Miao et al., 2021), GSM8K (Cobbe et al., 2021b), MATH-500 (Hendrycks et al., 2021b), AIME2024 (AI-MO, 2024a), and AMC (AI-MO, 2024b) in Table 1. To ensure the stability of the evaluation, we performed multiple samplings for each dataset and took the average accuracy. For GSM8K, MATH-500, and MinervaMath, we sampled each question 4 times and took the average accuracy of the 4 solutions. For AIME24 and AMC23, we sampled each problem 8 times and took the average accuracy of the 8 solutions. The token count was calculated using the corresponding tokenizer of the language model of Skythought[1] library.

**Baselines.**    We compared two types of baselines:

*Training-free Methods.* A direct and representative category of such approaches is the *prompt-based methods* paradigm. We compared our approach with the well-known prompt-based baselines in the community, including TALE-EP (Han et al., 2024), which requires the prompt to be as simple as possible, and ConciseCoT (Lee et al., 2025), which demands the use of the most concise CoT steps during step-by-step reasoning. Another widely adopted paradigm is the family of *model-merging based methods*, which leverages the rich knowledge from short CoT Instruct and the long CoT model for model fusion, aiming to achieve the shortest yet most effective reasoning process. We compared this approach with the Avg. Merging method used in Kimi-1.5 (Team et al., 2025; Wu et al., 2025a) and some advanced merging method, like Task-Arithmetic-Merging, Ties-Merging, Ties-Dare-Merging, discussed in Long2Short Technical Report (Wu et al., 2025a). We also compared with methods like Seal (Chen et al.), which extract hidden states, construct steer vectors, and use it to perform thought control during LLM decoding.

*Training-based Methods* In addition to prior work such as CoT-Valve and TOPS that constructs SFT datasets with diverse reasoning lengths, we also include approaches that incorporate alternative

---

[1] https://github.com/NovaSky-AI/SkyThought

| Model | Accuracy | | | | | | | Generation Length | | | | | | A.C.R. |
|---|---|---|---|---|---|---|---|---|---|---|---|---|---|---|
| | ASDiv | GSM8K | MATH | AIME | AMC | Minerva | Avg. | ASDiv | GSM8K | MATH | AIME | AMC | Minerva | |
| ***DS-Qwen-7B Models*** | | | | | | | | | | | | | | |
| R1-Distill-Qwen | 86.8 | 89.4 | 86.8 | 42.9 | 81.5 | 46.0 | 72.2 | 769 | 554 | 2861 | 6820 | 4510 | 3347 | – |
| TALE-EP | 80.4 | 89.1 | 84.3 | 40.0 | 80.0 | 42.3 | 69.3 | 509 | 450 | 1994 | 6520 | 3892 | 2242 | 22.3% |
| ConciseCoT | 86.0 | 89.5 | 86.2 | 41.7 | 79.6 | 46.0 | 71.5 | 532 | 457 | 2330 | 6587 | 4245 | 3347 | 12.7% |
| Avg. Merging | 92.8 | 70.1 | 58.6 | 0.05 | 39.6 | 29.8 | 48.4 | 622 | 8552 | 8540 | 8501 | 8542 | 8544 | 3.2% |
| Task-Arithmetic-Merging | 83.3 | 84.6 | 74.6 | 20.0 | 63.5 | 39.6 | 61.0 | 321 | 383 | 907 | 2500 | 1311 | 794 | 61.3% |
| Ties-Merging | 74.4 | 69.7 | 59.8 | 13.6 | 42.5 | 23.2 | 47.2 | 1114 | 2475 | 4086 | 6767 | 5195 | 4306 | 0.1% |
| Ties-Dare-Merging | 75.9 | 72.3 | 65.4 | 14.6 | 45.6 | 24.3 | 49.6 | 1036 | 2073 | 2934 | 5483 | 3698 | 2938 | 8.3% |
| Seal | 86.8 | 89.4 | 89.4 | 43.3 | 77.8 | 47.8 | 72.4 | 591 | 773 | 2661 | 6871 | 4740 | 3413 | 5.1% |
| Overthink | 86.6 | 89.6 | 87.2 | 38.7 | 79.6 | 45.2 | 71.1 | 773 | 555 | 2898 | 6766 | 4558 | 3407 | 0.1% |
| ThinkPrune | 90.6 | 92.1 | 91.0 | 43.3 | 86.2 | 45.6 | 74.8 | 653 | 587 | 2379 | 6207 | 3739 | 2762 | 12.6% |
| CoT-Valve* | 59.4 | 88.4 | 84.2 | 41.2 | 80.6 | 41.9 | 65.9 | 140 | 514 | 2144 | 6397 | 4278 | 2172 | 26.8% |
| TLDR | 93.0 | 87.7 | 87.4 | 41.2 | 83.1 | 41.0 | 72.3 | 147 | 253 | 1556 | 6368 | 3386 | 1451 | 44.9% |
| Δ | +6.2 | -1.7 | +0.6 | -1.7 | +1.6 | -5.0 | +0.1 | -622 | -301 | -1305 | -452 | -1124 | -1896 | – |
| ***DS-Qwen-14B Models*** | | | | | | | | | | | | | | |
| R1-Distill-Qwen | 80.5 | 92.5 | 86.4 | 43.4 | 79.6 | 48.2 | 71.7 | 476 | 679 | 2951 | 6701 | 4584 | 3270 | – |
| TALE-EP | 77.5 | 92.4 | 85.4 | 49.2 | 80.3 | 50.0 | 72.5 | 369 | 555 | 2248 | 6551 | 4179 | 2731 | 15.4% |
| ConciseCoT | 74.0 | 92.4 | 85.6 | 51.6 | 82.3 | 47.1 | 72.2 | 369 | 555 | 2066 | 6267 | 3878 | 2605 | 18.8% |
| Avg. Merging | 94.8 | 90.3 | 73.0 | 10.8 | 55.0 | 44.1 | 61.3 | 167 | 366 | 5158 | 6364 | 5668 | 1084 | 30.5% |
| Task-Arithmetic-Merging | 86.5 | 86.5 | 74.2 | 13.3 | 55.3 | 36.0 | 58.6 | 238 | 368 | 870 | 2813 | 1411 | 1050 | 60.2% |
| Ties-Merging | 79.6 | 91.3 | 82.6 | 25.4 | 72.5 | 37.1 | 64.8 | 242 | 542 | 1919 | 5913 | 3158 | 1850 | 31.8% |
| Ties-Dare-Merging | 80.7 | 91.8 | 84.8 | 25.4 | 75.3 | 34.9 | 65.4 | 274 | 467 | 1870 | 5747 | 3182 | 1877 | 33.0% |
| Overthink | 79.3 | 92.3 | 88.0 | 45.8 | 82.8 | 45.6 | 72.3 | 451 | 679 | 2893 | 6700 | 4464 | 3715 | 1.6% |
| ThinkPrune | 80.6 | 93.7 | 89.0 | 50.8 | 88.7 | 50.7 | 75.6 | 379 | 563 | 2177 | 5778 | 3327 | 2234 | 22.8% |
| CoT-Valve& | 72.9 | 92.0 | 87.0 | 45.0 | 83.5 | 47.8 | 71.4 | 204 | 576 | 2652 | 6686 | 4392 | 2833 | 16.7% |
| TLDR | 88.0 | 90.9 | 86.6 | 43.3 | 83.8 | 48.7 | 73.5 | 158 | 240 | 2092 | 6403 | 3839 | 2177 | 35.8% |
| Δ | +8.0 | -1.6 | +0.2 | -0.1 | +4.2 | +0.5 | +2.1 | -318 | -439 | -859 | -298 | -745 | -1093 | – |

Table 1: Performance comparison of TLDR with baselines. The accuracy is measured by sampling multiple responses from the LLMs and taking the average to reduce variance. * denotes the CoT-Valve result that we reproduced using the official dataset. Δ refers to TLDR in comparison with *Original*. Math′ and Minerva′ refer to MATH500 and MinervaMath datasets, respectively. A.C.R. means the token compression ratio computed by Eq. 16. In the table, yellow represents *prompt-based* methods; green highlights Merging-based methods; red indicates *SFT-based* and *RL-based* methods.

| Model | Accuracy | | | | | Generation Length | | | | |
|---|---|---|---|---|---|---|---|---|---|---|
| | GSM8K | MATH | AIME | AMC | Avg. | GSM8K | MATH | AIME | AMC | Avg. |
| ***7B Models*** | | | | | | | | | | |
| Original Model | 89.4 | 86.8 | 42.9 | 81.5 | 75.2 | 554 | 2861 | 6820 | 4510 | 3686 |
| ***Compression by SFT on Static Dataset*** | | | | | | | | | | |
| -Direct-Mixture | 87.1 | 84.8 | 39.7 | 73.1 | 71.2 | 236 | 1221 | 5322 | 2560 | 2335 |
| -MixChain-Z-GSM8K& | 88.4 | 84.2 | 41.2 | 80.6 | 73.6 | 514 | 2144 | 6397 | 4278 | 3333 |
| -TOPS | 85.9 | 89.4 | 43.3 | 77.8 | 74.1 | 336 | 2145 | 4378 | 7024 | 3471 |
| ***Compression by Simple Data Schedule*** | | | | | | | | | | |
| -Large->Small Curriculum | 86.9 | 83.0 | 39.5 | 76.8 | 71.5 | 231 | 1335 | 5684 | 2841 | 2523 |
| -Small->Large Curriculum | 84.0 | 82.0 | 41.2 | 80.0 | 71.8 | 266 | 1766 | 6729 | 3968 | 3182 |
| -Random Re-weight | 84.0 | 81.0 | 37.5 | 79.6 | 70.5 | 246 | 1416 | 5828 | 2925 | 2604 |
| ***Ours*** | | | | | | | | | | |
| -TLDR | 87.7 | 87.4 | 41.2 | 83.1 | 74.9 | 253 | 1556 | 6368 | 3386 | 2891 |

Table 2: Performance comparison after removing the MinervaMath column. Accuracy is measured using multiple sampling runs to reduce variance. & denotes CoT-Valve (Ma et al., 2025) and TOPS (Yang et al., 2025) reproduced results.

*reward-based methods.* ThinkPrune (Hou et al., 2025a) uses progressive compression of RL training length to improve the effectiveness of context utilization during exploration. $SimPO_{shortest}$ was introduced in Overthink (Chen et al., 2025) to adjust the effectiveness of the RL algorithm by length-guided RL training.

**Main Results** As shown in Table 1, We observe that *Prompt-based* methods exhibit relatively high instability and comparatively low compression ratios. *Merging-based* methods, on the other

| Training Method | Training Time (GPU hours) | Avg. Acc | A.C.R |
|---|---|---|---|
| *-T$_L$D$_R$* | ~88.1 | 72.3 | 44.9% |
| *-CoT-Valve* | ~240.6 | 65.9 | 26.8% |
| *-L1* | ~640.3 | – | – |
| *-ThinkPrune* | ~704.1 | 74.8 | 22.8% |

Table 3: Training time comparison (measured in GPU days) between T$_L$D$_R$ and the other baseline method. GPU hours are measured using A100 GPUs. We compared with L1, ThinkPrune, and CoT-Valve.

hand, are prone to significant performance drops and fail to achieve effective thinking compression across datasets of varying difficulty. Moreover, for certain base models, they cannot compress tokens effectively, leading to weight collapse. *Training-free based* methods like Seal (Chen et al.), which use control thinking by steering, can maintain relatively good performance across multiple datasets. However, they may be limited by the representational capacity of hidden state vectors, making it difficult to generalize to broader mathematical datasets, and their effectiveness in compressing reasoning is somewhat constrained.. Compared to the training-based algorithms in ThinkPrune and OverThink, TLDR demonstrates a higher average compression ratio A.C.R. This advantage is quite notable, especially on datasets such as ADSiv and GSM8K, which suffer overthinking issues.

To demonstrate the advantages of our dynamic re-weighting method, we carefully designed several types of baselines: (1) Static method using static compression SFT data with carefully crafted techniques to control reasoning length; MixChain-Z-GSM8K and TOPS are two Long2Short datasets proposed by CoT-Valve and TOPS in Table 2. Compared to carefully constructed thinking compression SFT datasets, such as CoT-Valve and TOPS, our method achieves superior compression performance. (2) Dynamic methods leveraging curriculum learning to find the optimal balance of System-1 and System-2 data, linearly adjusting their sampling probabilities over training steps; and (3) Re-weighting System-1/2 data randomly to explore better System-1/2 data ratios. In Table 2, We introduced other dynamic data balancing methods, such as curriculum learning, to construct the optimal mix of System-1 and System-2 data. For *Large->Small Curriculum*, as the number of training steps increases, the proportion of System-1 data is linearly decreased and the proportion of System-2 data is increased every fixed steps. *Small→Large Curriculum* follows the opposite strategy. In addition, we introduce Random Re-weight as another baseline. Every fixed steps, the sampling ratio of System-1 and System-2 data is reset to a random proportion.

We compare our approach with ThinkPrune, L1, and CoT-Valve, three different training-based methods. ThinkPrune and L1 use RL to guide sample generation within a set length or quota, requiring very long RL steps to reach the desired objectives, which leads to high training costs. The results in Table 3 show that TLDR, benefiting from the relatively fast training process of SFT and employing dynamic reweighting, achieves faster training compared to CoT-Valve, which which requires extensively using model interpolation to generate a large number of CoTs with diverse lengths. It is worth noting that methods represented by ThinkPrune perform RL while truncating the context length, encouraging the model to produce correct solutions within this limited context. However, this method requires many training steps to optimize its performance.

## 5.2 Ablation of Different System-1/2 Source

We also discovered that incorporating higher-difficulty CoT data into a short-long mixed dataset could effectively eliminate redundancies in CoT for their compressed version. However, direct mixing could lead to performance degradation. After introducing a dynamic ratio method, we found that flexibly adjusting the ratio could effectively maintain performance in Table 4. We categorized the sources of questions in the thinking compression data into three difficulty levels: *easy*, *medium*, and *hard*. *easy* questions are from GSM8K, *medium* questions are from the training set of MATH500, and *hard* questions are from the s1 prompt questions.

**Short CoT Compression Generalization Analysis of Easy-to-Hard.** We tested the construction of System-1 data, examining the composition of data from different thinking compression sources. Our experiments found that constructing data based on low-difficulty problems could significantly reduce the token count of high-difficulty problems while maintaining performance. We found that using lower-difficulty problems to construct thinking compression data for redundancy removal can further generalize to higher-difficulty problems.

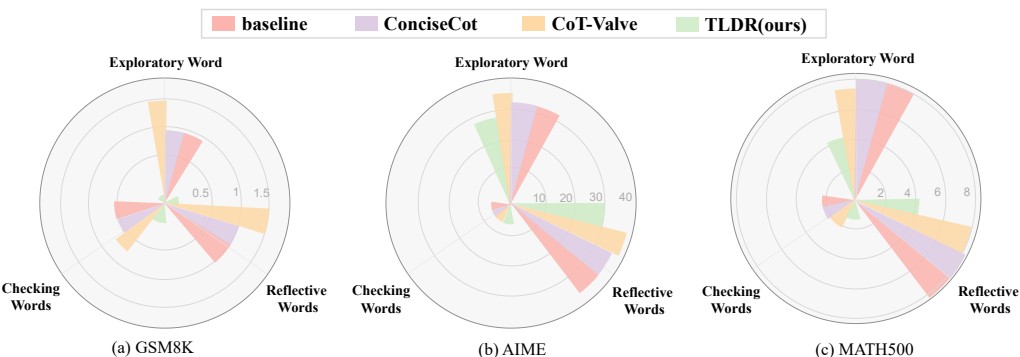

Figure 3: Frequency comparison of different keywords. The figure illustrates the distribution of exploratory, checking, and reflective keywords across datasets. *Exploratory Keywords: wait*, *Reflective Word: but*, *Checking Words: make sure/confirm/verify/check*, TLDR significantly reduces the presence of such words, reflecting its ability to produce streamlined and efficient reasoning steps.

| Model | Accuracy | | | | | Generation Length | | | | |
|---|---|---|---|---|---|---|---|---|---|---|
| | GSM8K | MATH | AIME | AMC | Avg. | GSM8K | MATH | AIME | AMC | Avg. |
| *DeepSeek-R1-Distill-Qwen-7B System-1 Short CoT Ablation* | | | | | | | | | | |
| Original Model | 89.4 | 86.8 | 42.9 | 81.5 | 60.1 | 554 | 2861 | 6820 | 4510 | 2949 |
| *-TLDR-Easy* | 87.7 | 87.4 | 41.2 | 83.1 | 59.9 | 253 | 1556 | 6368 | 3386 | 2313 |
| *-TLDR-Medium* | 88.2 | 86.2 | 41.5 | 31.3 | 61.8 | 318 | 2083 | 6604 | 3945 | 3238 |
| *-TLDR-Hard* | 83.6 | 80.2 | 30.0 | 65.3 | 64.8 | 495 | 2970 | 6874 | 4947 | 3822 |
| *DeepSeek-R1-Distill-Qwen-7B System-2 Long CoT Ablation* | | | | | | | | | | |
| Original Model | 89.4 | 86.8 | 42.9 | 81.5 | 60.1 | 554 | 2861 | 6820 | 4510 | 2949 |
| *-TLDR-Easy* | 83.9 | 86.8 | 42.5 | 83.4 | 74.2 | 446 | 2639 | 6580 | 4047 | 3428 |
| *-TLDR-Medium* | 91.6 | 87.6 | 40.4 | 81.5 | 75.3 | 542 | 2761 | 6553 | 4116 | 2950 |
| *-TLDR-Hard* | 87.7 | 87.4 | 41.2 | 83.1 | 74.8 | 253 | 1556 | 6368 | 3386 | 2828 |

Table 4: An ablation study on the difficulty levels of ShortCoT and LongCoT was conducted during the construction of the short CoT and long CoT dataset. The accuracy is measured by sampling multiple responses from the LLMs and taking the average to reduce variance. & denotes the CoT-Valve (Ma et al., 2025) result that we reproduced using the officially dataset. MATH means MATH500 dataset.

**Long CoT Performance Generalization Analysis of Hard-to-Easy.** We also conducted an analysis of the following aspects: during the sampling of long CoT, we utilized data from three distinct sources—*easy*, *medium*, *hard* prompt. Our findings reveal that only by constructing long CoT using hard problems and dynamically adjusting their proportions during training can we recover the original performance associated with long CoT. This strategy effectively mitigates the risk of forgetting in reasoning capabilities during continual learning.

## 5.3 COMPARISON WITH TOKEN BUDGETED-AWARE MODEL

We compared our redundancy reduction method with both quota-controlled models and reasoning models under the same token budget, in order to evaluate the effectiveness of our approach relative to explicit quota-based control in Table 5. The results show that our method achieves higher reasoning accuracy than both the L1 (Aggarwal & Welleck, 2025) baselines under the same token quota. Furthermore, our approach demonstrates more efficient utilization of context length and does not require explicitly specifying a reasoning quota, offering a more flexible and adaptive inference mechanism. TLDR demonstrates stronger compression efficiency on simple problems.

## 5.4 ANALYSIS OF THINKING PATTERNS: REFLECTIONS & SOLUTIONS

We compared our method with other thinking compression methods in terms of their impact on changes in cognitive patterns (Xu et al., 2025a) of the solution in Figure 3. We performed fine-grained statistical analysis on the results across different datasets. Our analysis demonstrates that our approach

| Model | Accuracy | | | | | Generation Length | | | | |
|---|---|---|---|---|---|---|---|---|---|---|
| | GSM8K | MATH | AIME | AMC | Avg. | GSM8K | MATH | AIME | AMC | Avg. |
| *DeepSeek-R1-Distill-Qwen-7B* | | | | | | | | | | |
| Original Model | 89.4 | 86.8 | 42.9 | 81.5 | 75.2 | 554 | 2861 | 6820 | 4510 | 3686 |
| -TLDR | 87.7 | 87.4 | 41.2 | 83.1 | 74.8 | 253 | 1556 | 6368 | 3386 | 2891 |
| -L1-same | 86.4 | 88.6 | 42.2 | 84.6 | 75.4 | 301 | 2301 | 5875 | 3784 | 3056 |
| -L1-lower | 86.4 | 87.6 | 45.1 | 84.6 | 75.9 | 312 | 1831 | 5675 | 3807 | 2906 |
| -L1-higher | 86.1 | 88.4 | 45.5 | 83.3 | 75.8 | 292 | 2589 | 6007 | 3746 | 3158 |

Table 5: Performance comparison of TLDR with budget-aware baseline, L1 (Aggarwal & Welleck, 2025). The accuracy is measured by sampling multiple responses from the LLMs and to reduce variance. The terms *same*, *lower*, and *higher* refer to setting the budget to match our results, 20% lower, and 20% higher, respectively. MATH means MATH500 dataset.

| Model | Leetcode | MBPP | HumanEval | Average |
|---|---|---|---|---|
| | Pass@1 / Tokens | Pass@1 / Tokens | Pass@1 / Tokens | Pass@1 / Tokens |
| *DS-7B* | 33.3 / 7088 | 61.4 / 1739 | 67.6 / 2692 | 54.1 / 3839 |
| DS-7B-*TLDR* | **34.4 / 6793** | **64.3 / 1234** | **73.1 / 2536** | **57.3 / 3521** |
| Δ | +1.1 / -295 | +2.9 / -505 | +5.5 / -156 | +3.2 / -318 |

| Model | ARC | HellaSwag | CommonsenseQA | Average |
|---|---|---|---|---|
| | Acc / Tokens | Acc / Tokens | Acc / Tokens | Acc / Tokens |
| *DS-7B* | 66.1 / 416 | 46.0 / 168.4 | 58.8 / 515 | 56.97 / 366.5 |
| *TLDR* | **68.1 / 488** | **47.5 / 180.1** | **59.5 / 495** | **58.37 / 387.7** |
| Δ | +2.0 / +72 | +1.5 / +11.7 | +0.7 / -20 | +1.40 / +21.2 |

Table 6: Performance comparison between *R1-Distill-Qwen-7B(DS-7B)* and *TLDR* across three coding and commonse QA benchmarks. Pass@1 accuracy is reported alongside the average number of tokens generated.

effectively compresses the internal redundancy and reflects the properties of the solution patterns. TLDR effectively reduces the reliance on such macro reasoning patterns in benchmarks like GSM8K and MATH500, thereby avoiding excessive allocation of computational budget. Notably, for more challenging problems, the model still retains a significant degree of complex reasoning behavior to preserve its System-2 reasoning capabilities, we provide additional case studies in the Appendix J.

### 5.5 DISCUSSION AND ANALYSIS ON THE NON-MATH DOMAIN BENCHMARK

Our training primarily relies on simple and challenging math problems from Math. We also aim to analyze and observe the model's generalization capabilities in non-math domains, including performance on CommonSenseQA and code reasoning tasks. Therefore, we selected HellaSwag (Zellers et al., 2019), CommonSenseQA (Talmor et al., 2019), and ARC (Clark et al., 2018) as three non-math domain datasets, and HumanEval (Chen et al., 2021), MBPP (Austin et al., 2021), and LeetCode (Guo et al., 2024) as three code-domain datasets. Since most questions in CommonSenseQA are multiple-choice and the original long CoT model's responses tend to be brief, we also incorporated some non-math subjects from MMLU (Hendrycks et al., 2021a) for comparison for reference in Appendix I.

## 6 CONCLUSION

This paper introduces TLDR, an innovative method designed to compress the reasoning processes of LLMs without sacrificing accuracy. By dynamically re-weighting the influence of System 1 (concise reasoning) and System 2 (detailed reasoning) data during the training process, TLDR allows LLMs to eliminate unnecessary steps for simpler problems while still engaging in deep contemplation for complex tasks. TLDR avoids the laborious data collection and hyperparameter tuning typically required by other compression methods, offering a more practical solution for developing LLMs that are both efficient and accurate.

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

APPENDIX CONTENTS FOR TLDR

## A USE OF LARGE LANGUAGE MODELS

During manuscript preparation, a large language model (LLM) was occasionally employed as an auxiliary assistant to refine language expression, such as improving sentence fluency and enhancing readability. The model was not involved in generating original research contributions: it did not participate in formulating research questions, designing methodologies, conducting experiments, analyzing results, or drafting substantive scientific content. All core intellectual work, including the development of ideas, execution of experiments, and interpretation of findings, was carried out independently by the authors. Any linguistic suggestions offered by the LLM were critically reviewed and selectively incorporated, ensuring that accuracy, originality, and scholarly integrity were fully maintained. The authors alone bear responsibility for the research content and conclusions, and the LLM is not listed as a contributor or author.

## B GRADIENT DERIVATION FOR EXPONENTIATED UPDATE OF $\alpha_i$

**Gradient-based Weight Updates of TLDR.** We consider the loss function:

$$L(\theta, \alpha) = \sum_{i=1}^{2} \alpha_i \cdot \delta_i, \quad \delta_i = \phi_{\text{sys-i,bound}} - \phi_{\text{sys-i},\theta}. \tag{9}$$

Assuming $\theta$ is fixed, $\delta_i$ can be treated as a constant. Thus, $L$ is linear in $\alpha_i$. $\alpha_i$ is required to be non-negative, and $\alpha_1 + \alpha_2 = 1$.

**Exponentiated Gradient (EG) Method.** The Exponentiated Gradient algorithm[2] is a standard technique in online convex optimization for updating probability-like weights under a convex loss function. Given a loss function $L(\alpha)$, the EG update is defined as:

$$\alpha_{t+1}[i] \leftarrow \alpha_t[i] \exp(-\eta \frac{\partial L}{\partial \alpha_i}), \tag{10}$$

where $\eta > 0$ is a learning rate. EG ensures that $\alpha_i > 0$, and after normalization, the weights form a valid probability distribution. It's particularly useful when weights represent probabilities or allocation ratios, as it avoids the need for explicit projection steps. Besides, it automatically maintains non-negativity and can be normalized to satisfy $\sum_i \alpha_i = 1$

**Exponentiated Gradient Update** The exponentiated gradient (EG) update for $\alpha_i$ is given by:

$$\alpha_{t+1}[i] \leftarrow \alpha'_t[i] = \alpha_t[i] \exp(-\eta \frac{\partial L}{\partial \alpha_i}) = \alpha_t[i] \exp(-\eta \delta_i), \tag{11}$$

where $\eta$ is the learning rate. This update increases the weight of components with smaller loss and decreases the weight of components with larger loss.

Due to potential variance or sampling errors in the validation set, we normalize $\delta_i$ and apply a thresholding operation to obtain a more stable estimate:

$$\lambda_i = \max\left(\frac{\delta_i}{\text{scale}_i}, 0\right),$$

where $\text{scale}_i$ is a normalization factor (the maximum of $\delta_i$), and $\lambda_i$ represents the normalized and non-negative estimate. The we get the result of

$$\lambda_{\text{sys-1}} = \max\left(\frac{\phi_{\text{sys-1, bound}} - \phi_{\text{sys-1},\theta_{\text{proxy}}}}{\phi_{\text{sys-1},\theta_s} - \phi_{\text{sys-1},\theta_l}}, 0\right) \tag{12}$$

$$\lambda_{\text{sys-2}} = \max\left(\frac{\phi_{\text{sys-2, bound}} - \phi_{\text{sys-2},\theta_{\text{proxy}}}}{\phi_{\text{sys-2},\theta_l} - \phi_{\text{sys-2},\theta_s}}, 0\right) \tag{13}$$

---

[2] https://homes.cs.washington.edu/~sham/courses/stat928/lectures/lecture22.pdf

**Normalization (considering $\sum_i \alpha_i = 1$ is desired)** To ensure that the weights remain a valid distribution, we normalize after the update:

$$\alpha_{t+1}[i] \leftarrow \frac{\alpha'_t[i]}{\sum_j \alpha'_t[j]} = \frac{\alpha_t[i]\exp(-\eta\delta_i)}{\sum_j \alpha_t[j]\exp(-\eta\delta_j)}. \tag{14}$$

Thus, formally, this is equivalent to treating $\lambda_i$ as a non-negative gradient signal and applying it in the exponentiated gradient update:

$$\alpha_{t+1}[i] = \alpha_t[i] \cdot \frac{\exp(-\eta \cdot \lambda_i)}{\sum_j \alpha_t[j]\exp(-\eta\lambda_j)}, \tag{15}$$

where $\eta$ is the learning rate.

## C  METRICS

### C.1  COMPRESSION RATE

We provide more details on the compression rate in the main table, where the compression rate is defined as:

$$\text{C.R.} = \text{Compression Rate} = max\left(\frac{\#\text{tokens}_{\text{original}} - \#\text{tokens}_{\text{current}}}{\#\text{tokens}_{\text{original}}}, 0\right) \tag{16}$$

$$\text{A.C.R.} = \frac{1}{N_{\text{benchmark}}} \sum_{i=0}^{N_{\text{benchmark}}} \text{C.R.} \tag{17}$$

### C.2  NORMALIZED METRIC

We report two normalized metrics to facilitate fair comparisons: Normalized Accuracy and Normalized Token Length. They are defined as follows:

$$\text{Normalized Accuracy} = \frac{\#\text{Acc}_{\text{current}}}{\#\text{Acc}_{\text{original}}} \tag{18}$$

$$\text{Normalized Token} = \frac{\#\text{Token}_{\text{current}}}{\#\text{Token}_{\text{original}}} \tag{19}$$

## D  DATA CONSTRUCTION DETAIL

For long CoT, we use the prompt from dataset s1.1 (Muennighoff et al., 2025). Each sample is generated 8 times using the original model. For short CoT, to avoid inconsistencies in the system prompt format, we adopt the short CoT construction method from AdaR1 (**?**). We annotate 10 randomly selected questions from GSM8K using the instruct model, then fine-tune the long CoT model to overfit on them. For the GSM8K training set, we sample and retain only the examples with correct answers.

## E  EVALUATION DETAIL

We use the DeepSeek-R1-Distill model and apply a temperature setting of 0.7, which is the primary recommendation in QwQ-Preview, for evaluating all models. All datasets are restricted to an 8K context window for output generation. Meanwhile, considering the relatively small sizes of the AMC and AIME datasets, we sample 8 responses per question and compute the average.

### E.1  EVALUATION FRAMEWORK

We use *skythought-eval*[3] as the framework, which supports accelerating long CoT reasoning evaluation with vLLM. The version of vLLM we use is 0.6.3.

---

[3]https://github.com/NovaSky-AI/SkyThought

## E.2 Evaluation Dataset Detail

We provide an overview of all datasets used in the following sections.

- **ASDiv**: A diverse simple English math word problem corpus for evaluating the capability of various MWP solvers. It contains 2,305 MWPs that cover more text patterns and most problem types taught in elementary school.
- **GSM8K**: A high-quality benchmark comprising 8,500 human-written grade school math word problems that require multi-step reasoning and basic arithmetic, each labeled with a natural language solution and verified answer. The 1,319-question test set emphasizes sequential reasoning and is primarily solvable by upper-grade elementary school students.
- **MATH500**: A challenging benchmark of 500 high school competition-level problems spanning seven subjects, including Algebra, Geometry, Number Theory, and Precalculus. Each problem is presented in natural language with LaTeX-formatted notation, offering a strong measure of mathematical reasoning and generalization across diverse topics.
- **AIME2024**: A dataset containing 30 problems from the 2024 American Invitational Mathematics Examination (AIME), a prestigious high school mathematics competition for top-performing students. Each problem is designed to require deep mathematical insight, multi-step reasoning, and precise problem-solving skills.
- **AMC**: The AMC dataset consists of all 83 problems from AMC12 2022 and AMC12 2023, extracted from the AoPS wiki page. We used a subset of this data containing 40 problems.
- **MinervaMath**: MinervaMath is a high-difficulty math problem dataset containing 272 challenging problems.

## F  Baseline Reproduce Details

**ConciseCoT & TALE-EP.** For the prompt-based baseline, we list the prompts used in Prompt 4.

**OverThink.** For the MATH12K dataset, we sample each problem 8 times. The shortest correct sample is selected as the chosen sample, and the longest sample is selected as the rejected sample. The model is trained for 1 epoch.

**ThinkPruner.** In our reproduction, we use the competition-level training data provided in the original paper and train the model for 10 epochs with a learning rate of 1e-6. The maximum response length is set to 4096 tokens. We follow their early stopping strategy to select the optimal checkpoint for evaluation.

**CoT-Valve.** Since CoT-Valve does not report performance on all datasets, we reproduced the results using the public datasets released by CoT-Valve. We followed the training settings officially reported in the paper, using LoRA=2 to fine-tune all models. The dataset version used is Mix-Chain-Z-GSM8K. All models were fine-tuned for 5 epochs on 8 GPUs with 80GB of memory each.

**L1.** In L1 reproduction on the 7B System-2 model, we utilize the *L1-Exact* reward function and limit the token length to between 100 and 4,096 tokens, while setting the token difference penalization parameter $\alpha$ to 0.0003, as described in the paper. We follow their original prompt by appending "Think for $n_{token}$ tokens" to the end of the question. In inference, the token budget is set to the same number as the average tokens from our method across the evaluated benchmarks.

## G  Training Details

Due to the need to evaluate accuracy and token count on a validation set every n steps, our validation set consists of 512 questions sampled from past questions in AIME-1983 to AIME-2023. The original ratio for shortcot and longcot is set to 0.5:0.5, with an evaluation interval of every 32 steps. The model is allowed to train for a total of $T$=500 steps, and the learning rate is set as a constant at 1e-5. For the 7B and 14B models, we conducted training on two 8-GPU (80GB) machines, with one 8-GPU machine performing vllm inference and the other performing training. Every n steps, parameter synchronization is executed using vllm's parameter sync function. We ultimately select the checkpoint with the shortest token length among those whose accuracy on the validation set is no less than 30% of that achieved by the original long CoT.

> **Evaluation Prompt on Dataset**
>
> **=== EVALUATION PROMPT FOR GSM8K ===**
> <|begin_of_sentence|>Please reason step by step, and put your final answer within 'boxed'.
> <|User|>query<|Assistant|>Given the following problem, reason and give a final answer to the problem.
> Problem: {question}
> Your response should end with Ïhe final answer is [answer]ẅhere [answer] is the response to the problem. <think>
>
> **=== EVALUATION PROMPT FOR MATH500 ===**
> <|begin_of_sentence|>Please reason step by step, and put your final answer within '
> boxed'. <|User|>{query}<|Assistant|>Return your final response within 'boxed'. {problem}. <think>
>
> **=== EVALUATION PROMPT FOR AIME24 ===**
> <|begin_of_sentence|>Please reason step by step, and put your final answer within '
> boxed'. <|User|>query<|Assistant|>Return your final response within '
> boxed'. {problem}. <think>
>
> **=== EVALUATION PROMPT FOR AMC ===**
> <|begin_of_sentence|>Please reason step by step, and put your final answer within 'boxed'.
> <|User|>query<|Assistant|>Return your final response within 'boxed'. {problem}. <think>
>
> **=== EVALUATION PROMPT FOR MINERVAMATH ===**
> <|begin_of_sentence|>Please reason step by step, and put your final answer within 'boxed'.
> <|User|>query<|Assistant|>Return your final response within 'boxed'. problem. <think>

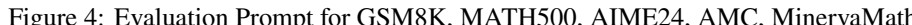

Figure 4: Evaluation Prompt for GSM8K, MATH500, AIME24, AMC, MinervaMath

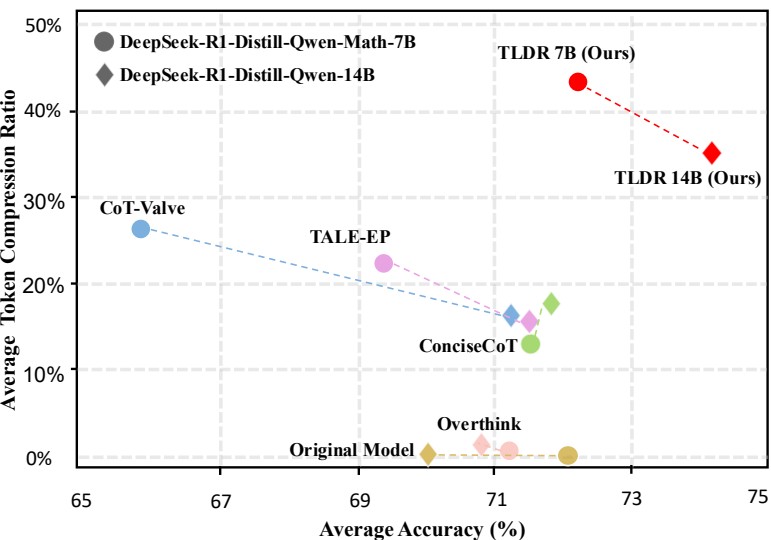

Figure 5: Comparison of TLDR and baseline models in terms of average accuracy and token compression ratio on 6 math benchmarks, including GSM8K, MATH, AIME24, and three others. Higher values on both axes indicate better performance.

# H    MORE RELATED WORK

Another category concentrates on building *efficient representations*. TokenSkip (Xia et al., 2025) selects data based on token importance for compressed reasoning and more concise thought chains. COCONUT (Hao et al., 2024) explores more efficient reasoning in the latent space. ICoT-KD (Cheng & Van Durme, 2024) and CCoT (Cheng & Van Durme, 2024) attempt to build more efficient reasoning

| Subject | Accuracy | | | Tokens | | |
|---|---|---|---|---|---|---|
| | DS-7B | TLDR-7B | Δ | Tokens | Tokens | Δ |
| *College Chemistry* | 54.0 | 60.0 | +6.0 | 3086.03 | 2489.37 | -596.66 |
| *Electrical Engineering* | 72.4 | 65.5 | -6.9 | 1553.04 | 968.78 | -584.26 |
| *Conceptual Physics* | 87.2 | 78.3 | -8.9 | 1110.13 | 806.51 | -303.62 |
| *College Computer Science* | 79.0 | 70.0 | -9.0 | 2732.12 | 1871.20 | -860.92 |

Table 7: Accuracy and token usage comparison between *DeepSeek-Distill-Qwen-7B(DS-7B)* and TLDR-7B across four subjects. Δ indicates the difference (*TLDR-7B - DS-7B*), with positive accuracy gains in green and token reductions in green.

strategies in the hidden space, while Token Assorted combines hidden space and text-based reasoning to balance interpretability and efficiency.

## I   MORE NON-MATH STEM REASON BENCHMARK IN MMLU

We also included additional non-mathematics STEM benchmarks in Table 7 to evaluate whether models trained primarily on mathematical reasoning can generalize to other STEM domains. Specifically, we assessed sub-tasks in MMLU, including college-level chemistry, conceptual physics, computer science, and electrical engineering. Our observations indicate that the model is capable of reducing the number of generated tokens in these tasks as well. For non-mathematical reasoning tasks, TLDR-7B exhibits a modest reduction in token count, likely attributable to the relative scarcity of training data in these domains.

## J   CASE STUDY

To better understand the behavioral differences between baseline and TLDR strategies, we conduct a qualitative analysis using the DeepSeek-R1-Distill-Qwen-7B model. Case studies are drawn from three representative math datasets: GSM8K, AIME, and MATH500. As shown in Figures 6–8, the baseline model tends to generate verbose reasoning paths with redundant or speculative content. In contrast, TLDR produces significantly more concise outputs while maintaining correctness and logical structure. These examples demonstrate TLDR's ability to suppress unnecessary reasoning tokens—such as exploratory or reflective phrases—leading to more efficient and focused reasoning processes.

## Generated Answer For GSM8K

**Question:**
Michael is traveling on the interstate at an average speed of 50km/hr (taking into account all the necessary stops for gas, relaxation, etc.). If he goes on like this for 3 days, and the distance between Alaska and Texas is 6000 km, what percentage of this distance has he covered?

### Baseline (842 tokens)

**Step 1: 114 tokens**

Okay, let me try to figure out how much of the distance from Alaska to Texas Michael has covered. So, Michael is driving at an average speed of 50 km per hour, and he's doing this for 3 days. The total distance between Alaska and Texas is 6000 km. I need to find out what percentage of that 6000 km he's covered.\n\nFirst, I think I should calculate the total distance Michael has driven in those 3 days. To do that, I need to know how many hours he's been driving each day. Hmm, the problem says he's taking into account all stops for gas, relaxation, etc., so I guess he's driving non-stop each day.

**Step 2: 87 tokens**

But wait, how many hours is that? Oh, the problem doesn't specify the number of hours he drives each day. That's a bit confusing. Maybe I'm supposed to assume a certain number of hours per day? Or perhaps the 50 km/h speed is his average speed including stops, so I can still use it to find the total distance.\n\nWait, maybe I can just calculate the total distance based on his average speed and the time. But without knowing the exact number of hours, I can't do that.

· · · · · ·

**Step 13: 40 tokens**

Alternatively, maybe the problem meant 3 hours each day, but that's unclear. Wait, perhaps the problem is in a different context, like driving across multiple states for 3 days, but without knowing the daily driving time, it's hard to say.

**Step 14: 49 tokens**

Maybe I should proceed with the initial assumption of 24 hours per day, even though it's unrealistic, and provide that answer.\n\nSo, final calculation: 50 km/h * 72 h = 3600 km. 3600/6000 = 0.6 => 60%.\n</think>\n\nThe final answer is 60%.

### TLDR(248 tokens)

**Step 1: 82 tokens**

Let's break down the problem step by step.
1. Calculate the total distance Michael will travel: Michael is traveling at an average speed of 50 km/hr for 3 days. To find the total distance, we multiply the speed by the number of days: Total distance = 50km/hr × 3 days = 150km.

**Step 2: 150 tokens**

2. Calculate the percentage of the distance between Alaska and Texas that Michael has covered: The distance between Alaska and Texas is 6000 km. To find the percentage of this distance that Michael has covered, we use the formula:
$$\text{Percentage} = \frac{\text{Distance covered}}{\text{Total distance}} \times 100$$
Substituting the values:
$$\text{Percentage} = \frac{150}{6000} \times 100 = 2.5$$

**Step 3: 16 tokens**

So, Michael has covered 2.5% of the distance between Alaska and Texas.

Figure 6: Comparison of Reasoning process on GSM8K: Baseline vs. TLDR.

## Generated Answer For AIME

**Question:**
Let B be the set of rectangular boxes with surface area 54 and volume 23. Let r be the radius of the smallest sphere that can contain each of the rectangular boxes that are elements of B. The value of $r^2$ can be written as $\frac{p}{q}$, where p and q are relatively prime positive integers. Find p + q

### Baseline (7482 tokens)

**Step 1: 87 tokens**

Observe that the \"worst\" possible box is one of the maximum possible length. \nBy symmetry, the height and the width are the same in this antioptimal box. (If the height and width weren't the same, the extra difference between them could be used to make the length longer.) Thus, let the width and height be of length $a$ and the length be $L$. \nWe're given that the volume is $23$; thus, $a^2L=23$.

**Step 2: 168 tokens**

We're also given that the surface area is $54=2\\cdot27$; thus, $a^2+2aL=27$. \nFrom the first equation, we can get $L=\\dfrac{23}{a^2}$. We do a bunch of algebra: \n\\begin{align*}\nL&=\\dfrac{23}{a^2} 27&=a^2+2aL \\\\\n&=a^2+2a\\left(\\dfrac{23}{a^2}\\right) \\\\\n&=a^2+\\dfrac{46}a \\\\27a&=a^3+46 \\\\na^3-27a+46&=0. \\\\\n\\end{align*}\nWe can use the Rational Root Theorem and test a few values. It turns out that $a=2$ works. We use synthetic division to divide by $a-2$: \n[Asdf.png](https://artofproblemsolving.com/wiki/index.php/File:Asdf.png)\nAs we expect, the remainder is $0$, and we are left with the polynomial $x^2+2x-23$.

......

**Step 27: 168 tokens**

To find the maximum m for $x^3-mx^2+27x-23=0$\nrewrite m as function of x and calculate derivatives to get maximum value, \\[m(x) =-x + 27x^{-1} - 23x^{-2}\\]\n\\[m'(x) = -1 - 27x^{-2} - 46x^{-3} = 0\\]\n\\[x^3 -27x+46=0\\]\n\\[(x-2)(x^2+2x-23)=0\\]\nwhen x = 2, \\[m= 2 + \\frac{27}{2}  - \\frac{23}{4}  = \\frac{39}{4}\\]\nthe rest is similar to solution 3\n~[luckuso](https://artofproblemsolving.comhttps://artofproblemsolving.com/wiki/index.php/User:Cyantist)

### TLDR(5960 tokens)

**Step 1: 84 tokens**

Observe that the \"worst\" possible box is one of the maximum possible length. \nBy symmetry, the height and the width are the same in this antioptimal box. (If the height and width weren't the same, the extra difference between them could be used to make the length longer.) Thus, let the width and height be of length $a$ and the length be $L$. \nWe're given that the volume is $23$; thus, $a^2L=23$.

**Step 2: 155 tokens**

We're also given that the surface area is $54=2\\cdot27$; thus, $a^2+2aL=27$. \nFrom the first equation, we can get $L=\\dfrac{23}{a^2}$. We do a bunch of algebra: \n\\begin{align*}\nL&=\\dfrac{23}{a^2} \\\\27&=a^2+2aL \\\\\n&=a^2+2a\\left(\\dfrac{23}{a^2}\\right) \\\\\n&=a^2+\\dfrac{46}a \\\\27a&=a^3+46 \\\\na^3-27a+46&=0. \\\\\n\\end{align*}\nWe can use the Rational Root Theorem and test a few values. It turns out that $a=2$ works. We use synthetic division to divide by $a-2$: \n[Asdf.png](https://artofproblemsolving.com/wiki/index.php/File:Asdf.png)\nAs we expect, the remainder is $0$, and we are left with the polynomial $x^2+2x-23$.

......

**Step 8: 60 tokens**

To find the maximum m for $x^3-mx^2+27x-23=0$\nrewrite m as function of x and calculate derivatives to get maximum value, \\[m(x) =-x + 27x^{-1} - 23x^{-2}\\]\n\\[m'(x) = -1 - 27x^{-2} -46x^{-3} = 0\\]\n\\[x^3 -27x+46=0\\]\n\\[(x-2)(x^2+2x-23)=0\\]\nwhen x = 2, \\[m= 2 + \\frac{27}{2}  - \\frac{23}{4} = \\frac{39}{4}\\]\nthe rest is similar to solution 3.

Figure 7: Comparison of Reasoning process on AIME: Baseline vs. TLDR.

## Generated Answer For MATH500

**Question:**
The set of points $(x,y,z)$ that satisfy\n\\[2x = 3y = -z\\]is a line.\n\nThe set of points $(x,y,z)$ that satisfy\n\\[6x = -y = -4z\\]is another line.\n\nFind the angle between these lines, in degrees.

### Baseline (1165 tokens)

**Step 1: 138 tokens**
Okay, so I've got this problem where there are two lines defined by sets of equations, and I need to find the angle between them. Hmm, let me see. The first line is given by the equations 2x = 3y = -z, and the second line is 6x = -y = -4z. I remember that to find the angle between two lines, I need to find the angle between their direction vectors. So, first step is probably to figure out the direction vectors of each line.\n\nLet me think about the first line: 2x = 3y = -z. I think that when an equation like this is given, it's setting all these expressions equal to some parameter, maybe t.

**Step 2: 237 tokens**
So, I can write each variable in terms of t. Let me try that. Let's let 2x = 3y = -z = t. Then, x would be t/2, y would be t/3, and z would be -t. So, the parametric equations for the first line would be x = t/2, y = t/3, z = -t. \n\nNow, the direction vector of a line given in parametric form is the coefficients of the parameter t. So, looking at x = (1/2)t, y = (1/3)t, z = (-1)t, the direction vector for the first line is (1/2, 1/3, -1). Hmm, that seems a bit messy with fractions. Maybe I can scale it up to make it simpler. If I multiply each component by 6, which is the least common multiple of denominators 2 and 3, I get (3, 2, -6). So, the direction vector for the first line is (3, 2, -6). That makes things easier.\n\nAlright, moving on to the second line: 6x = -y = -4z. Similarly, I can set all of these equal to another parameter, maybe s. So, 6x = -y = -4z = s.

· · · · · ·

**Step 10: 68 tokens**
3. Calculate the dot product of the direction vectors \\((3, 2, -6)\\) and \\((2, -12, -3)\\):\n   - \\(u \\cdot v = 3 \\cdot 2 + 2 \\cdot (-12) + (-6) \\cdot (-3) = 6 - 24 + 18 = 0\\).\n\nSince the dot product is zero, the vectors are perpendicular, meaning the angle between the lines is 90 degrees.\n\n\\[\n\\boxed{90}

### TLDR(204 tokens)

**Step 1: 44 tokens**
To find the angle between the two lines, we first need to determine their direction vectors. For the first line, given by \\(2x = 3y = -z\\), we can express it in parametric form. Let \\(2x = 3y = -z = t\\).

**Step 2: 52 tokens**
Then, we have:\n\n\\[\nx = \\frac{t}{2}, \\quad y = \\frac{t}{3}, \\quad z = -t\n\\]\n\nThus, the direction vector \\(\\mathbf{v}\\) of the first line is \\(\\left(\\frac{1}{2}, \\frac{1}{3}, -1\\right)\\).\n\nFor the second line, given by \\(6x = -y = -4z\\), we can express it in parametric form as well.

**Step 3: 37 tokens**
Let \\(6x = -y = -4z = s\\). Then, we have:\n\n\\[\nx = \\frac{s}{6}, \\quad y = -s, \\quad z = \\frac{s}{-4}\n\\]\n\nThus, the direction vector \\(\\mathbf{w}\\) of the second line is \\(\\left(\\frac{1}{6}, -1, -\\frac{1}{4}\\right)\\)

**Step 4: 45 tokens**
Next, we use the dot product formula to find the angle between the two direction vectors. The dot product \\(\\mathbf{v} \\cdot \\mathbf{w}\\) is calculated as follows:\n\n\\[\n\\mathbf{v} \\cdot \\mathbf{w} =\\left(\\frac{1}{2}\\right)\\left(\\frac{1}{6}\\right) + \\left(\\frac{1}{3}\\right)(-1) + (-1)\\left(-\\frac{1}{4}\\right) = \\frac{1}{12} - \\frac{1}{3} + \\frac{1}{4}\n\\]

**Step 5: 26 tokens**
Simplifying the expression: [\n\\frac{1}{12} - \\frac{4}{12} + \\frac{3}{12} = \\frac{0}{12} = 0\n\\]\n\nSince the dot product is zero, the vectors are perpendicular, and thus the angle between the lines is \\(90^\\circ\\).

Figure 8: Comparison of Reasoning process on MATH500: Baseline vs. TLDR.

