# OpenReview forum: "TL;DR: Too Long, Do Re-weighting for Efficient LLM Reasoning Compression"
_ICLR.cc/2026/Conference — ICLR 2026 Conference Withdrawn Submission_

### Official Review · Reviewer_oLZs · 2025-10-31

**Soundness:** 2
**Presentation:** 3
**Contribution:** 3
**Rating:** 4
**Confidence:** 3

**Summary:**

This paper addresses the issue of "over-deliberation" in Large Language Models (LLMs), where extensive System-2 (slow, deliberate) reasoning leads to excessive output token lengths and high inference costs, even on simple problems. The authors propose a novel training method called TLDR (Thinking Length Data Re-Weighting). Instead of relying on complex reinforcement learning or model merging, TLDR uses a dynamic SFT approach. It defines two data categories: System-1 (short CoT on easy problems) to encourage efficiency and System-2 (long CoT on difficult problems) to maintain reasoning accuracy. During training, the method dynamically adjusts the mixing ratio of these two data sources, guided by the model's performance on a validation set (monitoring both accuracy and output length). Experiments on 7B and 14B models show that TLDR can reduce output tokens by nearly 40% while maintaining accuracy comparable to the original, lengthy model, and is significantly more training-efficient than RL-based baselines.

**Strengths:**

Clear Motivation & Strong Rationale: The paper is well-motivated by a practical problem (inference efficiency). The analysis in Section 2, which shows that naively mixing data fails, provides a strong and clear justification for the necessity of the proposed dynamic re-weighting approach.

Method Simplicity and Novelty: The TLDR method is elegant. By reformulating the compression problem as a dynamic data-weighting task solved with SFT, it avoids the high complexity and instability of reward-based (RL) methods or the architectural issues of model merging.

Excellent Empirical Results: The method demonstrates significant, practical gains. It achieves a high token compression ratio (~40-45%) across multiple math benchmarks (Table 1) while suffering minimal to no accuracy degradation.

Training Efficiency: A key practical strength is the method's training speed. Table 3 shows TLDR is 8-10x faster than comparable RL-based methods like ThinkPrune or L1, making it a much more viable solution for model compression.

**Weaknesses:**

Limited Evaluation Domain: The experiments are conducted exclusively on mathematical reasoning datasets (GSM8K, MATH, AIME, etc.). It is unclear if this "System-1/System-2" data paradigm and the TLDR method will generalize to other reasoning domains, such as commonsense reasoning (e.g., HellaSwag), code generation, or long-form creative/factual writing.

Dependency on Curated "Hard" Data: The method's success seems to rely on the availability of a high-quality "difficult" dataset (like S1) to source the System-2 data. The sensitivity to the choice and quality of this dataset is not fully explored, and it may be a bottleneck for applying the method to new domains where such data is not readily available.

**Questions:**

Generalization to Other Tasks: How do you anticipate TLDR would perform on non-mathematical tasks? For example, in a domain like code generation, what would constitute "System-1" (easy) and "System-2" (hard) data, and would you expect a similar trade-off between efficiency and accuracy?

Sensitivity to Data Source: The ablation study (Table 4) shows that "hard" data is best for System-2. How sensitive is the method to this? For instance, what happens if the System-2 data is sourced from a "medium" difficulty dataset (like MATH500) instead of the "hard" S1 dataset? Does performance just degrade, or does the re-weighting algorithm fail to stabilize?

Baseline Comparison (L1): In Table 5, TLDR is compared to the L1 baseline under a "same" token budget. How was this budget determined? Is it the average token length of the final TLDR model, and was the L1 model trained specifically to target this exact budget?

I would like to improve my scores if authors can solve some questions.

---

> ### Author Response · Authors · 2025-11-23
> **Response to Reviewer oLZs (Part: 1/2)**
>
> We sincerely thank your valuable feedback. We greatly appreciate the time and effort spent in providing these insightful comments, and we provide point-by-point responses below.
>
> **>>> W1 & Q1**
>
> **>>>> “Limited Evaluation Domain: The experiments are conducted exclusively on mathematical reasoning datasets (GSM8K, MATH, AIME, etc.). It is unclear if this "System-1/System-2" data paradigm and the TLDR method will generalize to other reasoning domains, such as commonsense reasoning (e.g., HellaSwag), code generation, or long-form creative/factual writing.”**
>
> We appreciate your suggestions on evaluating TLDR on a broader range of downstream benchmarks, which could improve the comprehensiveness of our method.  We conducted comparisons on Code(Leetcode, HumanEval, MBPP)  and Commonsense Reasoning tasks (such as CommonseQA, ARC, HellaSwag and MMLU). We found that our model, trained on Math data, can effectively reduce redundant reasoning in Code tasks, mitigating overthinking and improving performance.
>
> | Model                       | Leetcode   | MBPP        | HumanEval   |
> |-----------------------------|------------|-------------|-------------|
> | DS-7B (R1-Distill-Qwen)     | 33.3/7088  | 61.4/1739   | 67.6/2692   |
> | DS-7B-TLDR                  | 34.4/6793  | 64.3/1234   | 73.1/2536   |
>
> On Commonsense QA datasets like HellaSwag, the original model did not exhibit significant overthinking; its outputs were generally concise.
>
> | Model                       | ARC       | HellaSwag    | CommonseQA |
> |-----------------------------|-----------|--------------|------------|
> | DS-7B (R1-Distill-Qwen)     | 66.1/416  | 46.0/168.4   | 58.8/515   |
> | DS-7B-TLDR                  | 68.1/488  | 47.5/180.1   | 59.5/495   |
>
> We also tested other STEM subjects beyond mathematical reasoning, utilizing sub-tasks in MMLU, such as college chemistry, conceptual physics, college computer science, and electrical engineering, and observed that our model could reduce the number of generated tokens in these tasks as well. For non-math reasoning tasks, TLDR-7B shows a slight reduction in token count, mainly due to the lack of training data in these domains.
>
> | Subject                   | DS_7B Acc | TLDR_7B Acc | Acc Delta  | DS_7B Token | TLDR_7B Token | Token Delta |
> |---------------------------|-----------|-------------|------------|-------------|---------------|-------------|
> | college chemistry         | 0.5400    | 0.6000      | +0.0600    | 3086.03     | 2489.37       | -596.66     |
> | electrical engineering    | 0.72414   | 0.65517     | -0.06897   | 1553.04     | 968.78        | -584.26     |
> | conceptual physics        | 0.87234   | 0.78298     | -0.08936   | 1110.13     | 806.51        | -303.62     |
> | college computer science  | 0.7900    | 0.7000      | -0.0900     | 2732.12     | 1871.20       | -860.92     |
>
> **>>> W2 & Q2**
>
> **>>>> “Dependency on Curated "Hard" Data: The method's success seems to rely on the availability of a high-quality "difficult" dataset (like S1) to source the System-2 data.
> The sensitivity to the choice and quality of this dataset is not fully explored, and it may be a bottleneck for applying the method to new domains where such data is not readily available”**
>
> We sincerely thank you for your insightful question. We greatly appreciate your careful reading of our work and the thoughtful feedback you have provided.
>
> ***Regarding the reviewers' concern about sourcing System-2 data from high-quality "difficult" datasets (such as S1)***
>
> There are alternative ways to label difficulty. S1 collects difficult problems by sampling multiple times with the model and selecting data with relatively low accuracy and longer length as difficult problems. We believe that it is possible to construct difficult problems in a similar way using questions and answers that are verifiable, like code reasoning or other similar QA Task.
>
> ***Regarding the sensitivity to the choice and quality of this dataset***
>
> Previously, we used MATH and GSM8K as substitutes for System-2 prompts to explore the impact of difficulty, but the results were not as good as the later S1. We are exploring the choice of difficulty and the quality of the dataset.
>
> We would like to clarify that we’ve included this part of experiments in our initial submission, and we kindly refer the reviewer to Table 4 for the results, where the “TLDR medium” corresponds to the results obtained by applying the System-2 algorithm on data constructed from medium-difficulty problems. We also included the simpler GSM8K dataset as “easy” problems to construct System-2 data for a comparison experiment. The results indicate that constructing System-2 data requires sufficiently difficult problems for effective training.
>
> However, for tasks such as long-form creative or factual writing, a better reward model or human filtering may be required.

---

> > ### Author Response · Authors · 2025-11-23
> > **Response to Reviewer oLZs (Part: 2/2)**
> >
> > **>>>  Q3**
> >
> > **>>>> Baseline Comparison (L1): In Table 5, TLDR is compared to the L1 baseline under a "same" token budget. How was this budget determined? Is it the average token length of the final TLDR model, and was the L1 model trained specifically to target this exact budget?**
> >
> > Yes, this corresponds to the average token length of the final TLDR model.
> >
> > At the inference stage, we set the budget to match the average token length of the final TLDR model for a fair comparison. Our approach, which does not require specifying a reasoning budget, can achieve accuracy that is comparable to L1 under the same token budget.
> >
> > As an RL-based approach, L1 requires a substantial amount of training time. In contrast, our method requires significantly less overall training time and additionally allows specifying an inference-time budget. As indicated in Table 3, it also appears to offer a reduction in training overhead.
> >
> > | Training Method | Training Time (GPU hours) | Avg. Acc | A.C.R  |
> > |-----------------|--------------------------|----------|--------|
> > | TLDR            | ~88.1                    | 72.3     | 44.9%  |
> > | CoT-Valve       | ~240.6                   | 65.9     | 26.8%  |
> > | L1              | ~640.3                   | –        | –      |
> > | ThinkPrune      | ~704.1                   | 74.8     | 22.8%  |

---

### Official Review · Reviewer_R9kL · 2025-10-31

**Soundness:** 3
**Presentation:** 1
**Contribution:** 2
**Rating:** 2
**Confidence:** 3

**Summary:**

The paper addresses the inefficiency of long Chain-of-Thought (CoT) reasoning in LLMs, which leads to excessive token usage during inference. It introduces **TL;DR**, a dynamic fine-tuning framework that adaptively balances short, intuitive (System-1) and long, deliberate (System-2) reasoning data using an exponentiated-gradient reweighting mechanism. This approach optimizes both efficiency and accuracy without requiring reinforcement learning or manual data curation. Experiments on multiple reasoning benchmarks show that TL;DR reduces output length while maintaining accuracy and training faster than prior RL-based compression methods.

**Strengths:**

* How to tune the length of reasoning is a well-known issue related to LLM performance. The paper addresses a key challenge in the area by proposing a new method.
* Empirically, the proposed method strikes a sweet balance between accuracy, inference reasoning length (or inference efficiency), and training efficiency (compared to RL).
* Concrete experiments include extensive ablation studies revealing multiple insights.

**Weaknesses:**

* The presentation is not clear, lacking citations. Many terms are not well defined.
  - In Line 40-41, the argument is not citing any prior work. It is not clear which work is the mainstream model merging that represents training-free methods.
  - In the paragraph starting from Line 73, the meaning of long CoT compression is not well defined. It is not clear if the proposed research is on training-based or training-free methods.
  - In Line 86-87, there is no citation for clarifying which GSM8k and s1 datasets are.
* Novelty is unclear, probably due to poor presentations. After reading the introduction, I am still confused about what is the techincal novelty of the method. Is this a data mixture method? What is the meaning of "re-weighting" in finetuning? What is the meaning of ratio learning? Why do we need a dynamic ratio? Since the reasoning length is associated with the difficulty of problems and the difficulties have been well defined in the datasets (GSM8k vs S1), a simple data pre-processing is enough to assign different length of reasoning to different problems of different difficulty.
* Missing related literature on the key finding. The trade-off between the short and long thinking was studied but not well reviewed in the paper. For example,  short thinking degrades performance due to unnecessary reasoning detours [A], and longer thinking can cause LLM to be trapped in redundant verification loops [B]. Admittedly, the effect of using short or long reasoning data in fine-tuning is worth studying, but the results are expected. When LLMs are trained on short reasoning, the LLMs will reason shorter and therefore based on [A], the LLM performance will be degraded.
* Because of the lack of related literature as aforementioned, the first contribution, the relation between reasoning length and accuracy, sounds trivial and its novelty is not well justified.
* Doubtable efficacy. In the main results, the comparison to training-based methos is not well discussed. In hard tasks like AIME, AMC, Minerva, the performance of the proposed method is consistently worse than ThinkPrune, even if it reduce more tokens. The gap in accuracy is not ignorable. For example, 50 -> 43 on AIME by DS model. Such gaps are not noted in the discussion, and it may undermine the claimed efficacy of the proposed method.

[A] Wang, Y., Liu, Q., Xu, J., Liang, T., Chen, X., He, Z., ... & Yu, D. (2025). Thoughts are all over the place: On the underthinking of o1-like llms. arXiv preprint arXiv:2501.18585.

[B] Chen, X., Xu, J., Liang, T., He, Z., Pang, J., Yu, D., ... & Yu, D. (2025). Do not think that much for 2+ 3=? on the overthinking of o1-like llms. ICML.

**Questions:**

Why are TOPS and other data synthesis work not included as a baseline in the experiments? Specifically, the authors mentioned that " Somework (Maetal.,2025;Jiangetal.,2025;Yuetal., 2025) synthesizes diverse-length CoT data, while TOPS (Yang et al., 2025) samples budget-sensitive versions using data model", showing that these works are closely related.

---

> ### Author Response · Authors · 2025-11-23
> **Response to Reviewer R9kL (Part: 1/4)**
>
> We sincerely thank the reviewers for their valuable feedback. We greatly appreciate the time and effort spent in providing these insightful comments, and we provide point-by-point responses below.
>
> **>>> W1**
>
> **>>>> "In Line 40-41, the argument is not citing any prior work. It is not clear which work is the mainstream model merging that represents training-free methods.“**
>
> To more clearly convey the substance and practical aspects of our method, we have expanded the Related Work section in edited version and moved it to Section 2, providing greater detail and addressing the gaps present in the earlier version. We sincerely hope that this will improve the readability of our work.
>
> As stated in lines 320–322 and 343–344, we compared the main Long2Short merging methods reported in technical reports, including the one used in Kimi-1.5 [1] as well as the Long2Short technical report [2].
>
> [1] Kimi k1. 5: Scaling Reinforcement Learning with
> LLMs. arXiv preprint arXiv:2501.12599, 2025
> [2] Unlocking efficient long-to-short llm reasoning with model
> merging, 2025. URL https://arxiv.org/abs/2503.20641
>
> **>>>> "In the paragraph starting from Line 73, the meaning of long CoT compression is not well defined. It is not clear if the proposed research is on training-based or training-free methods."**
>
> Long CoT Compression refers to a technique that compresses lengthy chain-of-thought outputs—especially those containing redundant self-correction, over-guarded reasoning styles, and repetitive validation loops—into a concise, information-dense format without losing essential reasoning steps.
>
> This concept has been widely discussed in the literature, such as training-based method CoT-Valve[3], ThinkPrune[4] and training-free model-merging method[5], as well as in training-free methods. TLDR is a training-based method, like ThinkPrune or CoT-Valve, that repeatedly adjusts the sampling ratio long CoT and short CoT before SFT.
>
> [3] CoT-Valve: Length-Compressible Chain-of-Thought Tuning.
>
> [4] ThinkPrune: Pruning Long Chain-of-Thought of LLMs via Reinforcement Learning.
>
> [5] Unlocking Efficient Long-to-Short LLM Reasoning with Model Merging
>
> We provide further explanations of the Long CoT Compression concept in the edited version of paper, aiming to improve the readability of the article.
>
> **>>>>  “In Line 86-87, there is no citation for clarifying which GSM8k and s1 datasets are.”**
>
> Thank you for your suggestion. We agree that providing these citations in the Introduction would improve the readability of the paper.
>
> Although we have already cited the relevant literature for the datasets in lines 154–158 and 307–308, we appreciate the reviewer’s recommendation and will also include these references in the Introduction to further enhance clarity and readability. Please refer to the revised version in our rebuttal. We sincerely welcome your other specific suggestions regarding further writing or presentation improvements.
>
> **>>> W2:**
>
> **>>>> “What is the meaning of "re-weighting”**
>
> Prior work has employed data re-weighting to adjust the proportions of data with varying quality and distribution during both pre-training and post-training, in order to achieve optimal training performance. We sincerely thank the reviewer for the valuable comments. We have added more related work in the Related Work section, and we hope this helps improve the readability of our paper.
>
> In our work,  “re-weighting” refers to dynamically adjusting the sampling weights of data from different sources (e.g., System-1 and System-2 data) during the SFT training process, thereby altering the probability of their selection in each training batch.
>
> **>>>> “What is the meaning of "ratio learning“**
>
> “ratio learning” refers to treating the sampling ratio between data categories (e.g., System-1 and System-2 data) as a learnable parameter during training, rather than a fixed preset. This ratio is dynamically determined by the optimization objective and updated via gradients.

---

> ### Author Response · Authors · 2025-11-23
> **Response to Reviewer R9kL (Part: 2/4)**
>
> **>>>> Why do we need a dynamic ratio? Since the reasoning length is associated with the difficulty of problems and the difficulties have been well defined in the datasets (GSM8k vs S1), a simple data pre-processing is enough to assign different length of reasoning to different problems of different difficulty.**
>
> Thank you for your insightful question. We would like to clarify our motivation again, and we hope this addresses your concern.
>
> Although in Section 2 we attempted a simple hybrid of System-1 and System-2 data to achieve an optimal balance between accuracy and sequence length, we found that this approach still resulted in some performance loss. To address this, we explored a dynamic method that actively seeks the best balance of data proportions.
>
> To further illustrate the importance of the dynamic approach and the necessity of using our method, we present three different baselines below. The first and second baseline uses a linear schedule to linear increase/decrease the proportion of System-1 and System-2 data. The second baseline, “random weight,” searches for the optimal System-1/System-2 data ratio through random trials. Approaches such as random trials or curriculum learning require a large number of experimental attempts, and unsuitable data proportions often make it difficult to maintain accuracy without loss.
>
> | Model                               | GSM8K      | MATH500      | AMC         | AIME        | Acc / Tokens |
> |-------------------------------------|------------|--------------|-------------|-------------|--------------|
> | DS-7B (R1-Distill-Qwen)             | 89.4/554   | 86.8/2861    | 81.5/4510   | 42.9/6820   | 75.1/3683    |
> | DS-7B-Curriculum-Large-to-Small*    | 86.9/231   | 83.0/1335    | 76.8/2841   | 39.5/5684   | 71.5/2522    |
> | DS-7B-Curriculum-Small-to-Large*    | 84.0/266   | 82.0/1766    | 80.0/3968   | 41.2/6729   | 71.8/3182    |
> | DS-7B-Random Weight*                | 84.0/246   | 81.0/1416    | 79.6/2925   | 37.5/5828   | 70.5/2603    |
> | DS-7B-TLDR                          | 87.7/253   | 87.4/1556    | 83.1/3386   | 41.2/6368   | 74.9/2890    |
>
> As for assign different length of reasoning to different problems of different difficulty.
>
> CoT-Valve and TOPS use this approach to construct reasoning datasets of varying lengths for each problem. However, this raises the question of how to determine the optimal CoT length for each problem. CoT-Valve and TOPS need to build multiple intermediate models (for example, TOPS first fine-tunes several tag models) to sample data, and generate a large amount of sampled data. This makes the process very cumbersome and requires a large volume of data. We compared the total SFT data used by CoT-Valve, TOPS, and TLDR as shown in the table below.
>
> | Method       | Number of CoTs Needed         | Require Intermediate Label Models? |
> |-------------|------------------------------|-----------------------------------|
> | TOPS        | 150K                         | Yes                               |
> | CoT-Valve   | 60K                          | Yes                               |
> | TLDR        | 7K (System-1) + 1K (System-2) | No                                |

---

> ### Author Response · Authors · 2025-11-23
> **Response to Reviewer R9kL (Part: 3/4)**
>
> **>>> W3 & W4:**
>
> **>>>> “Missing related literature on the key finding The trade-off between the short and long thinking was studied but not well reviewed in the paper. For example, short thinking degrades performance due to unnecessary reasoning detours [A], and longer thinking can cause LLM to be trapped in redundant verification loops [B]. Admittedly, the effect of using short or long reasoning data in fine-tuning is worth studying, but the results are expected.
>  When LLMs are trained on short reasoning, the LLMs will reason shorter and therefore based on [A], the LLM performance will be degraded.Because of the lack of related literature as aforementioned, the first contribution, the rel*atio*n between reasoning length and accuracy, sounds trivial and its novelty is not well justified.**
>
> References [A] and [B] were the first to identify the problems of overthinking and underthinking, and they each proposed a method—an RL-based method and a decoding-based method, respectively—to mitigate overthinking and underthinking.
>
> In our work, we first independently analyzed the impact of long-CoT and short-CoT data on model accuracy and reasoning length across tasks of varying difficulty. We identified features of data that are more favorable for optimizing length and accuracy and named these as **system-1 data** and **system-2 data**.
>
> Furthermore, we proposed a **dynamic reweighting strategy** that combines the two data types to search for the optimal trade-off between length and accuracy. Reference [A] was included in the corresponding baselines. We implemented the method **SimPO_{shortest}** proposed in [A] as a baseline.
>
> The results are reported in the table from the original submission at line 278 and line 288.
>
> Thank you for your feedback. We agree that adding a discussion of these two related works in the Section 2, together with the corresponding links and a clarification of how our experimental analysis differs from them, can improve the readability of our manuscript. Accordingly, we have revised Section 2 of the paper.
>
> **References:**
>
> [A] Wang, Y., Liu, Q., Xu, J., Liang, T., Chen, X., He, Z., ... & Yu, D. (2025). *Thoughts are all over the place: On the underthinking of o1-like LLMs.* arXiv preprint arXiv:2501.18585.
>
> [B] Chen, X., Xu, J., Liang, T., He, Z., Pang, J., Yu, D., ... & Yu, D. (2025). *Do not think that much for 2+3=? On the overthinking of o1-like LLMs.* ICML.
>
> **>>> W5**
>
> **>>>> “Doubtable efficacy. In the main results, the comparison to training-based methods is not well discussed. In hard tasks like AIME, AMC, Minerva, the performance of the proposed method is consistently worse than ThinkPrune, even if it reduce more tokens. The gap in accuracy is not ignorable. For example, 50 -> 43 on AIME by DS model. Such gaps are not noted in the discussion, and it may undermine the claimed efficacy of the proposed method.”**
>
> This is because **ThinkPrune** essentially encourages the model to truncate to the correct solution length while suppressing incorrect solutions and solutions that exceed the length. Implicitly, this method trains both **solution compression** and **reasoning ability** simultaneously.
>
> Although this approach can reduce solution length while improving reasoning ability, it is difficult to balance the two objectives reasonably, which leads to low training efficiency. Especially for relatively simple problems, all solution lengths fall below the specified truncation length. In such cases, the training objective focuses solely on reasoning ability.
>
> In **Table 3**, we compare the training time of our method with ThinkPrune. It can be seen that our method achieves better **training efficiency** while shows superior compression rate.
>
> | Training Method | Training Time (GPU hours) | Avg. Acc | A.C.R   |
> |-----------------|---------------------------|----------|---------|
> | TLDR            | ~88.1                     | 72.3     | 44.9%   |
> | CoT-Valve       | ~240.6                    | 65.9     | 26.8%   |
> | L1              | ~640.3                    | —        | –       |
> | ThinkPrune      | ~704.1                    | 74.8     | 22.8%   |

---

> > ### Author Response · Authors · 2025-11-23
> > **Response to Reviewer R9kL (Part: 4/4)**
> >
> > **>>> Q1: “Why are TOPS and other data synthesis work not included as a baseline in the experiments?”**
> >
> > We agree with your point and are very interested in comparing this baseline, as we believe it would further clarify and analyze the effectiveness of our method. However, the data used by **TOPS** was **not publicly available before the ICLR submission deadline**. We believe that including this baseline can improve the quality of the paper. We have now supplemented this experiment using the newly released data and compared it with **TLDR**.
> >
> > Here are the results of our additional comparison:
> >
> > Compared with TOPS, our method better preserves accuracy and also achieves improved compression. Moreover, as we mentioned earlier, by adopting the re-weight system-1/2 approach instead of sampling a large amount of data of varying lengths, we incur a relatively low data construction cost.
> >
> > | Model                  | ADSiV       | GSM8K      | MATH500      | AMC         | AIME        | MinervaMath   | Avg / C.R.   |
> > |------------------------|------------|------------|--------------|-------------|-------------|---------------|--------------|
> > | DS-7B (R1-Distill-Qwen)| 86.8/769   | 89.4/554   | 86.8/2861    | 81.5/4510   | 42.9/6820   | 46.0/3347     | 72.2/–       |
> > | DS-7B-TOPS             | 90.2/450   | 85.9/336   | 83.7/2145    | 78.9/4378   | 38.6/7024   | 42.1/1835     | 69.5/26.1%   |
> > | DS-7B-TLDR             | 93.0/147   | 87.7/253   | 87.4/1556    | 83.1/3386   | 41.2/6368   | 44.5/1451     | 72.8/44.9%   |

---

> > > ### Comment · Reviewer_R9kL · 2025-11-27
> > > **Thanks**
> > >
> > > Thank you for the rebuttal, which addressed most of my concerns.
> > > I agree that the paper proposed a novel method to dynamically re-weight different data lengths and effectively improved the reasoning ability.

---

> > > > ### Author Response · Authors · 2025-11-28
> > > > **Thank you for your reply**
> > > >
> > > > Thank you very much for your reply. We are glad that we could address your concern and appreciate your insightful suggestions.

---

### Official Review · Reviewer_TPrS · 2025-11-01

**Soundness:** 3
**Presentation:** 3
**Contribution:** 2
**Rating:** 6
**Confidence:** 3

**Summary:**

This paper presents TL;DR (Thinking-Length Data Re-weighting), a method for compressing the reasoning length of Large Language Models (LLMs) without degrading accuracy. The approach dynamically adjusts the ratio between System-1 (short CoT) and System-2 (long CoT) data, using feedback from token length and accuracy metrics. Experiments on DeepSeek-R1-Distilled Qwen models (7B/14B) across GSM8K, MATH-500, AIME, AMC, and Minerva show around 40% token reduction with negligible accuracy loss compared to baseline reasoning compression methods such as CoT-Valve, ThinkPrune, and L1.

**Strengths:**

- Compressing reasoning length for improved efficiency is an important and timely research direction, and the proposed method demonstrates strong empirical effectiveness.
- The paper includes extensive ablation studies, providing thorough and convincing evaluations.
- The proposed approach is clearly presented, easy to follow, and supported by released code, enhancing reproducibility.

**Weaknesses:**

- I am somewhat unconvinced about the necessity of the dynamic re-weighting strategy in Algorithm 1. An ablation study comparing different re-weighting strategies would help clarify its contribution. For example, including simple baselines such as fixed curriculum ratios (large-to-small, small-to-large, or random re-weighting) could provide a clearer understanding of the proposed method’s effectiveness.

- It would be helpful to clarify the inference settings in the experimental setup. For instance, the DeepSeek report uses a maximum sequence length of 32k and a temperature of 0.6 for AIME. From my own experience, AIME responses can easily exceed 10k tokens, which may not align with the results reported in Table 1.

**Questions:**

- How about the generalizaton of this method, current evlaution tasks are all math tasks, if we'd like to go beyond to other tasks like code or common-sense, will it be important to collect mixed data from different domain for training.

---

> ### Author Response · Authors · 2025-11-23
> **Response to Reviewer TPrS  (Part: 1/2)**
>
> We sincerely thank your valuable and constructive suggestions on both the writing and experimental aspects of our manuscript. We have carefully addressed these comments and revised the content accordingly. An updated and edited version of the manuscript has now been submitted.  Below, we address each point in detail through additional experiments and clarifications, and we greatly appreciate your time and careful evaluation.
>
> **>>> W1**
>
> **>>>> “I am somewhat unconvinced about the necessity of the dynamic re-weighting strategy in Algorithm 1, An ablation study comparing different re-weighting strategies would help clarify its contribution. For example, including simple baselines such as fixed curriculum ratios (large-to-small, small-to-large, or random re-weighting) could provide a clearer understanding of the proposed method’s effectiveness.”**
>
> We sincerely thank the reviewers for their suggestions. In addition to the static datasets, we have included additional baselines for curriculum learning as well as baselines with random reweighting.
>
> | Model                               | GSM8K      | MATH500      | AMC         | AIME        | Acc / Tokens |
> |-------------------------------------|------------|--------------|-------------|-------------|--------------|
> | DS-7B (R1-Distill-Qwen)             | 89.4/554   | 86.8/2861    | 81.5/4510   | 42.9/6820   | 75.1/3683    |
> | DS-7B-Curriculum-Large-to-Small*    | 86.9/231   | 83.0/1335    | 76.8/2841   | 39.5/5684   | 71.5/2522    |
> | DS-7B-Curriculum-Small-to-Large*    | 84.0/266   | 82.0/1766    | 80.0/3968   | 41.2/6729   | 71.8/3182    |
> | DS-7B-Random Weight*                | 84.0/246   | 81.0/1416    | 79.6/2925   | 37.5/5828   | 70.5/2603    |
> | DS-7B-TLDR                          | 87.7/253   | 87.4/1556    | 83.1/3386   | 41.2/6368   | 74.9/2890    |
>
> DS-7B-Curriculum-Large-to-Small and DS-7B-Curriculum-Small-to-Large are two relatively simple approaches applied to the System-1 and System-2 datasets, under the constraint of maintaining the same 500 training steps as TLDR.
>
> **DS-7B-Curriculum-Large-to-Small:** The ratio of System-1 to System-2 data is adjusted every 100 steps from the original 1:0 to 0.8:0.2, 0.6:0.4, 0.4:0.6, 0.2:0.8, and finally 0:1.
>
> **DS-7B-Curriculum-Small-to-Large:** The ratio of System-1 to System-2 data is adjusted every 100 steps from the original 0:1 to 0.2:0.8, 0.4:0.6, 0.6:0.4, 0.8:0.2, and finally 1:0.
>
> **DS-7B-Random Weight:** A new random ratio of System-1 to System-2 data is generated every 20 steps.
>
> The results indicate that using a purely random approach or relatively simple curriculum learning strategies makes it difficult to achieve an ideal trade-off, often leading to a decrease in accuracy.
>
> **>>> W2**
>
> **>>>> “It would be helpful to clarify inference settings in the experimental setup. For instance, the DeepSeek report uses a maximum sequence length of 32k and a temperature of 0.6 for AIME. From my own experience, AIME responses can easily exceed 10k tokens, which may not align with the results reported in Table 1.”**
>
> Considering the relatively small sizes of AIME datasets, we sample 8 responses per question and compute the average. Since some samples can reach the full 32K token length, causing vllm inference to be very slow, we conduct all our evaluation using an 8K context window. We also provide results for a context length of 32k, which we hope can help resolve this concern.
>
> | Model                       | ASDiv      | GSM8K      | MATH500      | AMC         | AIME-32K        | MinveraMath      | Avg./A.C.R |
> |-----------------------------|------------|------------|--------------|-------------|-----------------|------------------|------------|
> | DS-7B (R1-Distill-Qwen)     | 86.8/769   | 89.4/554   | 86.8/2861    | 81.5/4510   | 55.0 / 12951    | 46.0 / 3347      | 74.2/—     |
> | DS-7B-TLDR                  | 93.0/147   | 87.7/253   | 87.4/1556    | 83.1/3386   | 52.9 / 11908    | 44.5 / 1451      | 74.7/45.1% |

---

> > ### Author Response · Authors · 2025-11-23
> > **Response to Reviewer TPrS (Part: 2/2)**
> >
> > **>>> Q1**
> >
> > **>>>>> “How about the generalizaton of this method, current evlaution tasks are all math tasks, if we'd like to go beyond to other tasks like code or common-sense, will it be important to collect mixed data from different domain for training.”**
> >
> > We appreciate your suggestions. We conducted  zero-shot inference our model in other domain,  comparisons on Code (Leetcode, HumanEval, MBPP)  and Commonsense Reasoning tasks (such as CommonseQA, ARC, HellaSwag and MMLU).
> >
> > We found that our model, trained on math data, can effectively reduce redundant reasoning in Code tasks, mitigating overthinking and improving performance.
> >
> > | Model                       | Leetcode   | MBPP        | HumanEval   |
> > |-----------------------------|------------|-------------|-------------|
> > | DS-7B (R1-Distill-Qwen)     | 33.3/7088  | 61.4/1739   | 67.6/2692   |
> > | DS-7B-TLDR                  | 34.4/6793  | 64.3/1234   | 73.1/2536   |
> >
> > On Commonsense QA datasets like HellaSwag, the original model did not exhibit significant overthinking; its outputs were generally concise.
> >
> > | Model                       | ARC       | HellaSwag    | CommonseQA |
> > |-----------------------------|-----------|--------------|------------|
> > | DS-7B (R1-Distill-Qwen)     | 66.1/416  | 46.0/168.4   | 58.8/515   |
> > | DS-7B-TLDR                  | 68.1/488  | 47.5/180.1   | 59.5/495   |
> >
> > We also tested non-math tasks in MMLU, such as college chemistry, conceptual physics, college computer science, and college physics, and observed that our model could reduce the number of generated tokens in these tasks as well. For non-math reasoning tasks, TLDR-7B shows a slight reduction in token count, mainly due to the lack of training data in these domains.
> >
> > | Subject                   | DS-7B Acc | TLDR-7B Acc | Acc Delta  | DS-7B Token | TLDR-7B Token | Token Delta |
> > |---------------------------|-----------|-------------|------------|-------------|---------------|-------------|
> > | college chemistry         | 0.5400    | 0.6000      | +0.0600    | 3086.03     | 2489.37       | -596.66     |
> > | electrical engineering    | 0.72414   | 0.65517     | -0.06897   | 1553.04     | 968.78        | -584.26     |
> > | conceptual physics        | 0.87234   | 0.78298     | -0.08936   | 1110.13     | 806.51        | -303.62     |
> > | college computer science  | 0.7900    | 0.7000      | -0.0900     | 2732.12     | 1871.20       | -860.92     |
> >
> > We believe that collecting more domain-specific data should help improve performance and facilitate generalization to these domains.

---

### Official Review · Reviewer_FBQD · 2025-11-01

**Soundness:** 3
**Presentation:** 2
**Contribution:** 2
**Rating:** 4
**Confidence:** 3

**Summary:**

The paper proposes, TLDR: a mixed data construction and re-weighting of short and long CoT data for reasoning model's supervised finetuning. This helps the finetuned model generated precise yet accurate reasoning responses. The authors compared with prompt based, model merging based and RL based methods as baselines to demonstrate the efficacy of TLDR. The authors further categorize the sources
of questions in the thinking compression data into three difficulty levels: easy, medium, and hard, to perform short and long CoT performance generalization analysis.

**Strengths:**

1. The data mixing idea to make models generate concise yet correct reasoning traces is new to me.
2. The experiments were done on multiple datasets and models
3. The model performance is consistently similar or better.

**Weaknesses:**

1. The paper needs to be proof-read by a native english speaker, as there are certain grammar issues. Example: "on reasoning LLMs, enabling the model to learn to generate more concise $\textbf{yet still}$ correct reasoning paths."-- use "yet".

2. The authors did not compare with training free reasoning compression methods ([1-2]).

3. The overhead of manual selection easy to hard data as well as system-1, 2 categorization has additional labeling overhead in their SFT process.

[1] SEAL: Steerable Reasoning Calibration of Large Language Models for Free, COLM 2025.

[2] Activation Steering for Chain-of-Thought Compression, Arxiv 2025.

**Questions:**

1. Is there any training compute or latency overhead of the process?

2. Please compare with training free steering method for reasoning trace control.

---

> ### Author Response · Authors · 2025-11-23
> **Response to Reviewer FBQD**
>
> We sincerely thank the reviewers for their valuable and constructive suggestions on both the writing and experimental aspects of our manuscript. We have carefully addressed these comments and revised the content accordingly. An updated and edited version of the manuscript has now been submitted. We address each point in detail through additional experiments and clarifications, and we greatly appreciate your time and careful evaluation.
>
> **>>> W1: The paper needs to be proof-read by a native english speaker, as there are certain grammar issues**
>
> We sincerely thank the reviewers for their suggestions. We have revised the manuscript according to the points raised by the reviewers. We would greatly appreciate any further detailed suggestions for improvement.
>
> **>>> W2 & Q2: “The authors did not compare with training free reasoning compression methods ([1-2])”. “Please compare with training free steering method for reasoning trace control.”**
>
> We sincerely thank the reviewers for their suggestions, which offered a highly constructive baseline and valuable guidance for our research.
>
> In the main experiments of our paper, we compared training-free methods based on model merging, training-free methods via prompts. References [1–2] propose leveraging hidden states to steer the model to generate more efficient and concise chain-of-thought (CoT) reasoning.
>
> To highlight the differences between these approaches and our method, we reproduced the methods from the open-source repository “ZJU-REAL/EasySteer” to evaluate their inference compression capabilities on DS-7B. The results are shown below: our method achieves overall superior compression across more benchmarks while maintaining accuracy.
>
> [1] SEAL: Steerable Reasoning Calibration of Large Language Models for Free, COLM 2025.
> [2] Activation Steering for Chain-of-Thought Compression, arXiv 2025.
>
> | Model              | ASDiv        | GSM8K        | MATH500      | AMC          | AIME         | MinervaMath  | Avg / A.C.R.   |
> |--------------------|--------------|--------------|--------------|--------------|--------------|--------------|--------------|
> | DS-7B (R1-Distill-Qwen) | 86.8 / 769   | 89.4 / 554   | 86.8 / 2861  | 81.5 / 4510  | 42.9 / 6820  | 46.0 / 3347  | 72.2 / –     |
> | DS-7B-Seal*        | 86.8 / 591   | 89.4 / 773   | 89.4 / 2255  | 77.8 / 4740  | 43.3 / 6871  | 47.8 / 3413  | 72.4 / 15.8% |
> | DS-7B-TLDR         | 93.0 / 147   | 87.7 / 253   | 87.4 / 1556  | 83.1 / 3386  | 41.2 / 6368  | 44.5 / 1451  | 72.8 / 44.9% |
>
> **>>> W3 & Q1: “The overhead of manual selection easy to hard data as well as system-1, 2 categorization has additional labeling overhead in their SFT process.” “Is there any training compute or latency overhead of the process?”**
>
> Thank you very much for raising such an insightful question. We will further discuss the issues regarding data construction and the additional computational cost, and we hope to address your concern.
>
> Our approach mainly requires identifying some relatively difficult problem data.  A large amount of open-source data with standardized difficulty levels is already available.
>
> For datasets that lack explicit difficulty annotations but provide ground-truth answers, we can leverage model-based evaluation to distinguish between easy and hard problems. For example, the hard problem dataset we use, S1, is a model-constructed hard problem dataset, primarily created by generating reasoning samples based on the accuracy of reasoning models such as R1/QwQ on mathematical problems.
>
> Our approach requires only a small computational overhead during the sampling stage. We recorded the time spent on SFT data sampled from hard and easy problems, as well as their proportion in the overall TLDR training time. It can be seen that this portion consistently accounts for only a small part of the TLDR training.
>
> | Progress-TLDR-7B                | GPU A100 Hours | Percentage of Total |
> |---------------------------------|----------------|----------------------|
> | Data Sampling of System-1/2     | 10.1           | 10.28%               |
> | Dynamic Re-weight Training      | 88.1           | 89.72%               |
> | **Total = Sampling + Training** | **98.2**       | **100%**             |

---

### Note · Authors · 2026-01-06

I have read and agree with the venue's withdrawal policy on behalf of myself and my co-authors.